# Augmenting cross-entropy with margin loss and applying moving average logits regularization to enhance adversarial robustness

## Abstract

Despite significant progress in enhancing adversarial robustness, achieving a satisfactory level remains elusive, with a notable gap persisting between natural and adversarial accuracy. Recent studies have focused on mitigating inherent vulnerabilities in deep neural networks (DNNs) by augmenting existing methodologies with additional data or reweighting strategies. However, most reweighting strategies often perform poorly against stronger attacks, and generating additional data often entails increased computational demands. Our work proposes an enhancement strategy that complements the cross-entropy loss with a margin-based loss for generating adversarial samples used in training and in the training loss function of promising methodologies. We suggest regularizing the training process by minimizing the discrepancy between the Exponential Moving Average (EMA) of adversarial and natural logits. Additionally, we introduce a novel training objective called Logits Moving Average Adversarial Training (LMA-AT). Our experimental results demonstrate the efficacy of our proposed method, which achieves a more favorable balance between natural and adversarial accuracy, thereby reducing the disparity between the two.

## 1 Introduction

Our reliance on technology continues to grow, as evidenced by the undeniable progress in three essential computer vision tasks: object detection, face recognition, and image segmentation. Despite these advancements, deep neural networks (DNNs) (He et al., 2016b; Huang et al., 2017; Zagoruyko & Komodakis, 2016b; Szegedy et al., 2016) remain vulnerable to adversarial examples (Goodfellow et al., 2014; Szegedy et al., 2013; Yin et al., 2022; Mu et al., 2023). These adversarial examples are carefully crafted versions of the original input that appear visually identical to natural examples but can drastically mislead the model with high confidence (Athalye et al., 2018; Qin et al., 2019). Ensuring the robustness and adaptability of deployed models to diverse input perturbations is therefore crucial. In response to the vulnerability of DNNs, two primary approaches have emerged: adversarial detection and adversarial defense. Adversarial detection aims to identify malicious samples before they are fed to the model (Li & Li, 2017; Feinman et al., 2017; Xu et al., 2018). Adversarial defense, on the other hand, can be classified into two subgroups: certified and empirical defenses. Certified defenses (Cohen et al., 2019; Zhang et al., 2020a; Kumar & Narayan, 2022) aim to provide a provable guarantee of adversarial robustness to norm-bounded attacks. Empirical defenses have shown significant progress, particularly adversarial training (AT) (Goodfellow et al., 2015). Various variants have been proposed, including those by (Madry et al., 2018; Zhang et al., 2019; Wang et al., 2020; Ding et al., 2020; Wang et al., 2020; Fakorede et al., 2023a; Xie et al., 2020; Atsague et al., 2021; 2023), and (Li et al., 2021). More details on existing works in section 2.2. Formally, (Madry et al., 2018) formulated the adversarial training procedure as a min-max optimization problem, aiming to find the optimal network parameters $\theta$ that minimize the following risk:

$$\min_{\theta} \frac{1}{n} \sum_{i=1}^{n} l(f_{\theta}(x_i'), y_i), \tag{1}$$

where $l(.)$ is a loss function, $f_\theta(x_i)$ is the prediction of the neural network with parameters $\theta$ given an input $x_i$, and $y_i$ is the class label. In (1), the standard adversarial training (AT)(Madry et al., 2018) generates the adversarial example $x_i'$ using $x_i' = arg \max_{x' \in B_\epsilon[x_i]} g_i'(f_\theta(x'), y_i)$, which are then used to train the model. $g_i'(.)$ is the loss used to generate adversary examples , and $B_\epsilon[x] = \{x' \mid \|x' - x\|_p < \epsilon\}$ is a neighborhood of $x$. While the cross-entropy loss is widely used for generating adversarial examples, alternative methods exist. For example, in the loss function $g_i'(.)$, TRADES (Zhang et al., 2019) adopts the Kullback-Leibler Divergence. On the other hand, FAT (Zhang et al., 2020b) considers the cross-entropy but employs a misclassification-aware criterion, hence generating adversarial using $x_i' = arg \max_{x' \in B_\epsilon[x_i]} g_i'(f_\theta(x'), y_i)$ $s.t.$ $g_i'(f_\theta(x'), y_i) - \min_{y \in Y} g_i'(f_\theta(x'), y) \geq \rho$ where $\rho > 0$ is a margin such that adversarial data are misclassified with a certain amount of confidence. The objective in generating adversarial examples is to find the worst-case input, also known as the optimal adversarial example $x' \in B_\epsilon[x_i]$. Searching for the optimal adversarial used for training can be done in multiple ways; our work adopts the projected gradient descent (PGD) (Madry et al., 2018). Assuming a starting point $x^{(0)}$ referring to natural data perturbed by a small Gaussian or Uniformly random noise, i.e., $x^{(0)} = x_i + Gaussian/Uniform$ and is in the input feature space with distance metric $\|x - x'\|_\infty$. Let $t \in \mathbb{N}$. PGD generates adversarial examples using the following update rule:

$$x^{(t+1)} = \prod_{B[x_i]} (x^{(t)} + \alpha \cdot sign(\nabla_{x^{(t)}} g_i'(f_\theta(x^{(t)}), y_i))) \tag{2}$$

In (2), $\alpha$ is a step size, $\prod_{B[x_i]}(.)$ is the projection function, $x^{(t)}$ is the adversarial example at step $t$, and $g_i'(.)$ is the loss used to generate the adversarial used for training. In this work, $g_i'(.) = CE(.) + L(.)$ where $L(.)$ is a margin-based loss (more details in Section 4.2). Certain studies focus on refining loss functions and regularization techniques within the spectrum of adversarial training. Some of these methods aim to reduce the disparity between the output probabilities of adversarial examples and their corresponding natural counterparts. However, this strategy can hinder the learning process, especially if a natural example is misclassified (Dong et al., 2023). Despite the promising results of adversarial training and its variations, a significant gap remains between the natural and adversarial accuracy. Recent approaches have focused on refining existing methodologies to further enhance model performance. These improvements include perturbing network weights (Wu et al., 2020), weighting losses during training (Zhang et al., 2020c), and augmenting datasets with unlabeled and/or additional labeled data (Carmon et al., 2019; Zhai et al., 2019; Alayrac et al., 2019), among other strategies. Other approaches (Izmailov et al., 2018) explore model weight-averaging. In this approach, the weights are computed using the exponential moving average of the model parameters ($\theta' \leftarrow \tau * \theta' + (1 - \tau) * \theta$), where the parameter $\theta'$ replaces the model parameter $\theta$ during evaluation time. (Gowal et al., 2020) discovered that model weight averaging can significantly enhance robustness across different models and datasets. Inspired by their observation, we hypothesize that averaging the logits could enhance adversarial robustness. Hence, a regularization technique was introduced aimed at minimizing the disparity between natural and adversarial examples through the averaging of logits (more details in section 4.3). Extensive experiments demonstrate that we can build a more robust model by minimizing the disparity between the moving average of natural and adversarial logits. Many classification tasks widely adopt the Softmax function, which has also been used intensively in the adversarial machine-learning context, mainly due to its simplicity and probabilistic interpretation. Together with the cross-entropy loss, they form arguably one of the most commonly used components in CNN architectures (Liu et al., 2016).

We explored the adversarial class predictions using a ResNet-18 model trained on CIFAR-10. For this investigation, the adversarial examples were generated using the PGD-20 method, and the cross-entropy loss was employed for both training and adversarial data generation. For each input pair $(x_i, x_i')$ where $x_i$ and $x_i'$ are the natural and adversarial examples, respectively, we assume the second through tenth positions, in order, the most probable incorrect classes when $x_i$ is classified by a model trained under regular training. If $x_i'$ is wrongly classified, we track the class to which it is wrongly classified; it could be wrong classified to the 2nd, 3rd, ..., or the 10th most probable false class when $x_i$ is classified under normal training. We consider both PGD-20 and CW attacks to fool the model and record our findings in Fig 1, which indicates that when wrongly classified, most adversarial examples are wrongly classified into the 2nd, the 3rd, then the 4th, and 5th most probable false classes. A similar observation was made in (Li et al., 2021). Based on this observation, (Li et al., 2021) introduced a novel training objective called Probabilistically Compact (PC) loss with logit constraints to enhance adversarial robustness. However, a drawback of this approach is that

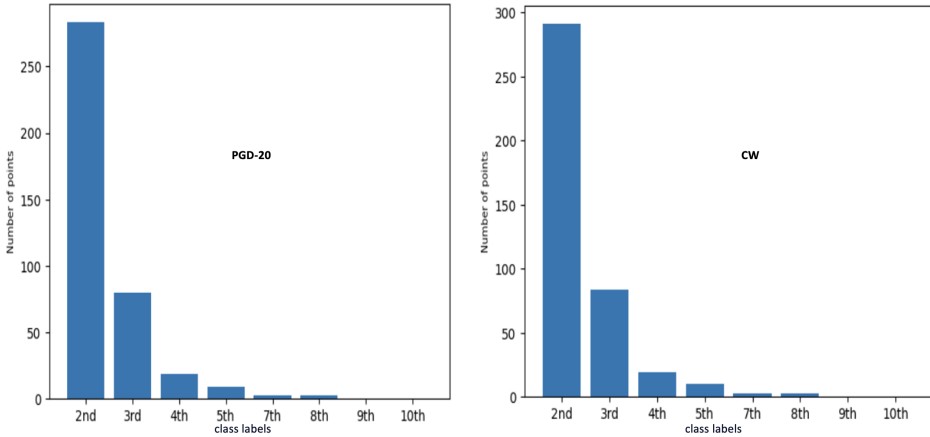

Figure 1: 2nd, 3rd, 4th,....correspond to the order of the most probable false classes under normal training (with natural input). The plot indicates the frequency of adversarial data points incorrectly classified in each class by the model trained on adversarial with the CE loss.

it entirely replaces the cross-entropy (CE) loss with the margin-based loss. It is not designed to compete with adversarial training methods but rather to be combined with them to improve robustness further. Consequently, this method requires further exploration to achieve true adversarial robustness compared to other promising adversarial training approaches. While the CE loss primarily focuses on the probability that the input is assigned to its ground-truth class without placing constraints on other class probabilities, we hypothesize that both maximizing the probability gaps between the actual class and the most probable false classes and ensuring that the input is correctly classified are crucial for improving model robustness against adversarial inputs. The phenomena in Fig 1 is expected. We postulate that the most probable misclassified classes are those that share the most features with the input. For instance, a model trained to recognize a jaguar might mistakenly classify it as a cheetah or leopard during an attack, as these species have many overlapping characteristics. This issue arises from the close proximity of predicted class probabilities. By incorporating a margin-based loss into our improvement strategy, we widen the separation between classes in the feature space, making it more difficult for adversarial attacks with small perturbations to mislead the model. Therefore, we need to maximize the probability gaps between the true and most probable false classes and increase the likelihood that the perturbed/natural input is classified correctly. In summary, we aim to satisfy the following conditions.

1. Maximize the adversarial probability gaps between the true and most probable false classes.

2. Increase the probability that the perturbed/natural input is assigned to its ground-truth class.

To satisfy the aforementioned criteria, we augmented the cross-entropy loss with a margin-based loss. Our experiments suggest that integrating these criteria into the generation of adversarial examples during training enhances the model's resilience against adversarial attacks. Consequently, we combined the cross-entropy loss with the margin-based loss for generating adversarial examples used in training. Furthermore, including the moving average of logits in the regularization process further enhances model performance. Our experiments illustrate that these techniques improve the model's ability to generalize on clean data while maintaining robustness against adversarial examples, notably narrowing the accuracy gap between natural and adversarial samples. Empirically, we demonstrate that this strategy effectively defends against common attacks and achieves a more favorable balance between natural accuracy and adversarial robustness. Our main contributions are summarized as follows:

* Unlike previous methods (Kannan et al., 2018; Atsague et al., 2021; 2023) that regulated training by focusing on natural and adversarial logits, we are pioneering a new approach. In our method, we incorporate both the current logits and those from the previous iteration, utilizing

a moving average calculation. This enables us to capture valuable comparative insights from the early stages of training.

* We augmented the cross-entropy loss with a margin-based loss, applying this approach both in generating the adversarial samples for training (inner maximization) and in the outer minimization to complement existing training losses. Building on enhancement strategies from previous research, we introduce a streamlined and highly efficient training objective called Logits Moving Average Adversarial Training (LMA-AT).

* Through rigorous experimentation, we validate that our proposed method consistently enhances the adversarial robustness of state-of-the-art techniques by a significant margin in certain attack scenarios.

## 2 Existing works

The vulnerability of deep learning has attracted significant attention, prompting efforts to mitigate this issue. These efforts include generating complex adversarial examples, developing defensive techniques, and establishing evaluation methodologies.

### 2.1 Adversarial attacks

A spectrum of attacks has been proposed to assess machine learning vulnerability and can be classified into two main categories: White-box attacks and Black-box attacks
**White-box attacks**: This list includes the Fast Gradient Sign Method (FGSM)(Goodfellow et al., 2014), which generates adversarial examples with a single normalized gradient step. It exploits the gradient sign at every pixel to determine which direction to change the corresponding pixel value. This attack is fast and simple; hence, it can be easily implemented. On the other hand, Projected Gradient Descent (PGD) (Madry et al., 2018) introduces a random starting point at each iteration in FGSM within a specified $l_\infty$ norm-ball to intensify the attack effect. In other words, it is an optimization procedure used to search norm-bounded perturbations. CW attack (Carlini & Wagner, 2017) consists of finding adversarial perturbations by introducing auxiliary variables which incorporate the pixel value constraint. In addition, we have Fast-Minimum-Norm (FMN) Attack (Pintor et al., 2021). FMN iteratively finds the sample misclassified with maximum confidence within an $l_p$-norm constraint of size $\epsilon$, while adapting $\epsilon$ to minimize the distance of the current sample to the decision boundary.
**Black-box attacks**: This list includes SQUARE attack (Andriushchenko et al., 2020), which is based on the randomized search scheme, does not rely on the local gradient information, and thus is unaffected by gradient masking. Hence, SQUARE attack is one of the best Black box attack assessment approaches. Along the same line, SPSA attack (Uesato et al., 2018) is a gradient-free method that approximates gradient to generate adversarial. In addition to commonly used attacks (SQUARE and SPSA), other black box attacks exist (Chen et al., 2017; 2020; Chen & Gu, 2020; Ma et al., 2021; Shukla et al., 2021).
**AutoAttack** (Croce & Hein, 2020b) combines both black-box and white-box attacks. It is an ensemble of parameter-free attacks that combine two parameter-free versions of PGD, APGD-CE (Croce & Hein, 2020b), and APGD-T (Croce & Hein, 2020b), with two existing complementary attacks, FAB-T (Croce & Hein, 2020a) and SQUARE attack.

### 2.2 Adversarial defenses

Various defensive techniques have emerged to bolster model robustness against adversarial attacks, categorized into certified and empirical defenses. Empirical defense considered the most successful approach, integrates adversarial data into the training process (Madry et al., 2018; Kannan et al., 2018; Cai et al., 2018; Zhang et al., 2019; Wang et al., 2019; 2020; Ding et al., 2020; Atsague et al., 2021; Rice et al., 2020; Atsague et al., 2023). To further enhance adversarial robustness, contemporary works incorporate extra unlabeled data (Carmon et al., 2019; Deng et al., 2021; Rebuffi et al., 2021); some incorporate synthetic data (Gowal et al., 2021; Wang et al., 2023). For example, (Sehwag et al., 2022) leverages additional data from

proxy distributions learned by advanced generative models. Another research direction explores reweighting (Liu et al., 2021; Zhang et al., 2020c; Fakorede et al., 2023b), where the training samples are treated unequally. As a result, various reweighting schemes have been proposed to assign different weights to the robust losses of individual examples in the training set based on specific conditions. Conversely, some researchers suggest that a single model lacks the capability to defend against all possible adversarial attacks, resulting in suboptimal robustness. Consequently, an emerging line of research has focused on developing ensembles of neural networks to enhance defense against adversarial attacks (Sen et al., 2020; Pang et al., 2019; Zhang et al., 2022). Our work aligns with existing efforts to improve adversarial robustness but significantly diverges from data augmentation, ensembling, and reweighting techniques. While reweighting shows promise against specific attacks, it performs poorly against stronger ones. We do not add additional data or incorporate a reweighting strategy on specific loss components of benchmark adversarial training to enhance adversarial robustness. Instead, we introduce a margin loss to constrain the probability that a data point is not assigned to its true class (further elaborated in Section 4). Additionally, we regularized the training by minimizing the disparity between the moving averages of the natural and adversarial logits. Before delving into our enhancement strategy, let's briefly discuss benchmark adversarial training approaches. (Madry et al., 2018) employ the standard cross-entropy loss. Adversarial Logit Pairing (ALP) (Kannan et al., 2018) introduces a regularization term that minimizes the mean square error loss between two logits (natural and adversarial logits). MIMAE-AT (Atsague et al., 2021) proposes two regularization terms: the mutual information between the probabilistic predictions of the natural example and its adversarial version, and the mean absolute error between their logits. TRADES (Zhang et al., 2019) theoretically characterizes the trade-off between accuracy and robustness of classification problems and suggests a regularization term that balances adversarial robustness against accuracy. Conversely, instead of enhancing adversarial training using a set perturbation magnitude, Max-Margin Adversarial (MMA) training (Ding et al., 2020) rethinks adversarial robustness through a margin-focused lens. It advocates for "direct" input margin maximization, aiming to maximize the margins computed for each data point to achieve optimal robustness. On the other hand, MART (Wang et al., 2020) introduces a regularization term that explicitly distinguishes between misclassified and correctly classified examples. WAT (Zeng et al., 2021) Proposed a formulation that considers the importance of the weights of different adversarial examples and focuses adaptively on examples that are wrongly classified or at higher risk of being classified incorrectly. Under this formulation, If the margin of the generated adversarial example during training $x'_{training}$ is large, the adversarial example $x'_{training}$ is considered a weak attack, and thus its importance weight should be smaller. A persistent limitation in existing works is the well-known trade-off between accuracy on clean images and adversarial robustness, as mentioned in (Tsipras et al., 2018). Additionally, methods that rely on additional data often increase the computational costs. However, our proposed method does not require extra data, making it suitable for resource-limited tasks. To address this issue, we supplemented the cross-entropy loss with the margin loss. We hypothesize that integrating the margin loss with cross-entropy loss can enhance robustness against adversarial attacks. The margin loss promotes greater class separation, helping the model resist adversarial perturbations, while the cross-entropy loss ensures accurate classification. This combination may improve robustness without significantly compromising the natural accuracy, as observed in the experimental results. Additionally, regularizing with the mHuber loss between the natural and adversarial moving averages of logits stabilizes the training by mitigating the effect of outliers and reducing the variance between natural and adversarial logits. As observed in the experimental section, our method improves robust accuracy while compromising less on the model's performance on clean samples than existing approaches, effectively addressing the trade-off between natural and adversarial accuracy better than other methods. Among the methods mentioned, our enhancement strategy is most compatible with Vanilla AT (Madry et al., 2018), TRADES (Zhang et al., 2019), and MART (Wang et al., 2020). Therefore, these methods will serve as the baseline for improvement.

## 3    Notations and preliminaries

Consider a classification problem over the data set $D = \{(x_i, y_i)\}_{i=1}^n$ where $x_i$ is a natural input example associated with the label $y_i \in Y = \{1, ......, C\}$ where $C$ is the number of classes. Let $f_c(x_i, \theta)$ be the *logit* output of the deep neural network with model parameters $\theta$ corresponding to class $c$ and $p_c(x_i, \theta) =$

$e^{f_c(x_i,\theta)}/\sum_{c'=1}^{C}e^{f_{c'}(x_i,\theta)}$ represent the probability that the network predicts class $c$ given the input example $x_i$. Let $f_\theta(x_i)$ represent the class prediction of the network. We denote by $l(.)$ and $E[l(.)]$ the loss and expected loss, respectively. The loss of the network over the dataset $D$ is defined by

$$E[l(.)] = \frac{1}{n}\sum_{i=1}^{n}l(f_\theta(x_i), y_i). \tag{3}$$

**mHuber Loss:** As defined in (Atsague et al., 2023), consider two vectors $u = [u_1, ..., u_n]$ and $v = [v_1, ..., v_n]$. The element-wise subtraction is $u - v = [u_1 - v_1, ..., u_n - v_n]$, and $|u - v| = [|u_1 - v_1|, ..., |u_n - v_n|]$. Let $c = [c_1, .., c_n]$ such that $c_i$ is True if $|u_i - v_i|/\alpha \leq \pi/2$, and False otherwise. In addition, let $A = [A_1, ..., A_n] = \alpha^2(1 - cos((u - v)/\alpha))$, $B = [B_1, ..., B_n] = \alpha|u - v| + (1 - \frac{\pi}{2})\alpha^2$, and $H = [H_1, ..., H_n]$ such that $H_i = A_i$ if $c_i$ is True, and $H_i = B_i$ if $c_i$ is False. Then $mHuber(u, v, \alpha) = mean(H) \equiv (H_1 + ... + H_n)/n$.

In the formulation, above, $A_i = \alpha^2(1 - cos((u_i - v_i)/\alpha))$, $B_i = \alpha|u_i - v_i| + (1 - \frac{\pi}{2})\alpha^2$, and $c_i$ determine whether $B_i$ or $A_i$ is applied based on the condition $|u_i - v_i|/\alpha \leq \pi/2$. In adversarial training, minimizing errors between logits as a regularization term is widely used, with mean absolute error (MAE) and mean square error (MSE) being popular choices in the literature. Each of these methods has its advantages and drawbacks. The MSE is highly sensitive to outliers, which can result in unpredictable outcomes (Liano, 1996), whereas the MAE is more robust to outliers but lacks differentiability at zero. The mHuber loss function's smooth second derivative makes it particularly suitable for scenarios where stability during training is crucial. The smooth second-order derivative improves robustness to outliers and noisy data, as demonstrated in (Guo et al., 2021). By mitigating the instability issues associated with the standard Huber loss, the mHuber loss ensures that gradient-based optimization methods can proceed more reliable, resulting in a more stable and potentially more accurate model. This can be especially important in applications where small changes in the errors can lead to large impacts on the final model performance, such as in high-precision regression tasks or in adversarial settings where robustness is critical. When applied to logits, The cosine term is applied to slight logit differences, which smooths the loss function and ensures a continuous second derivative. Which helps avoid the instability issues of the standard Huber loss.

## 4 Proposed Defense Method

### 4.1 Empirical Risk Formulation

There are inherent risks associated with inadequately trained models. A properly designed and trained model should accurately classify natural and adversarial inputs. Therefore, minimizing the risks associated with misclassifying both natural and adversarial inputs is imperative. To reduce the natural risk across the dataset $D$, we aim to minimize

$$Risk_{nat}(f_\theta(.)) = \frac{1}{n}\sum_{i=1}^{n}\mathbb{1}(f_\theta(x_i) \neq y_i), \tag{4}$$

where $\mathbb{1}(.)$ is the indicator function. When it comes to the adversarial risk, we consider the adversarial risk formulation of (Madry et al., 2018; Zhang et al., 2019) on the classifier $f_\theta(.)$ with the 0-1 loss over the dataset $D = \{(x_i, y_i)\}_{i=1}^{n}$ formulated as

$$Risk_{adv}(f_\theta(.)) = \frac{1}{n}\sum_{i=1}^{n}\max_{x_i' \in B_\epsilon[x_i]}\mathbb{1}(f_\theta(x_i') \neq y_i), \tag{5}$$

Most existing works (Madry et al., 2018; Zhang et al., 2019; Wang et al., 2020; Atsague et al., 2021; 2023) minimized the adversarial Risk in Equation 5. The problem with the risk formulation above is that they only care that the adversarial input needs to be assigned to the correct class and neglect how the assignment is done. The finding of Fig. 1 indicates that when the adversarial examples are wrongly classified, most are wrongly classified in the 2nd, 3rd, 4th, and 5th most probable false classes when classifying under normal training. Given the clean input pair $(x_i, y_i)$, let $S_p = \{p_j(x_i, \theta)\}_{j=1}^{C}$ represent the set of class probabilities when predicting under natural training, and $P_i(x_i, \theta) = max(S_p)$ represents the predicted class probability.

Let $y_k$ represent the $2nd$, $3rd$, $4th$, $5th$, ..., or the $qth$ most probable false classes, where $q < |C|$. Instead of minimizing the risk $Risk_{adv}$ defined in Equation 5, we constrain the adversarial risk to the following formulation:

$$Risk_{adv}(f_\theta(.)) = \frac{1}{n} \sum_{i=1}^{n} \max_{x_i' \in B_\epsilon[x_i]} \mathbb{1}(f_\theta(x_i') = y_k); \tag{6}$$

Where $y_k$ represents the $2nd$, $3rd$, $4th$, $5th$, ..., or the $qth$ most probable false classes. Given our goal of enhancing the most promising adversarial training methods, we focus on providing a risk formulation that aligns with our improvement strategy. We consider Vanilla AT (Madry et al., 2018), TRADES (Zhang et al., 2019), and MART (Wang et al., 2020). In the latter two methods, a regularization term minimizes $\mathbb{1}(f_\theta(x'i) \neq f_\theta(x_i))$, promoting consistency in classification decisions between natural and adversarial examples. Our objective is for the model to accurately classify both types of examples. Hence, minimizing the risk of misclassifying natural and adversarial examples is crucial. In conclusion, our improvement strategy for Vanilla AT, MART, and TRADES involves minimizing both $Risk_{adv}$ in Equation 6 and $Risk_{nat}$ in Equation 4.

### 4.2 Surrogate losses

Directly minimizing the empirical risks $Risk_{nat}(f_\theta(.))$ in Equation 4, $Risk_{adv}(f_\theta(.))$ in Equation 6 and $\mathbb{1}(f_\theta(x_i') \neq f_\theta(x_i))$ with 0-1 loss is intractable. An appropriate convex surrogate loss usually replaces the 0-1 loss. TRADES (Zhang et al., 2019) minimizes the natural risk (Equation 4) in which the $\mathbb{1}(f_\theta(x_i) \neq y_i)$ term is replaced by the cross-entropy (CE) loss. However, TRADES does not explicitly minimize the adversarial risk defined in Equation 6. On the other hand, the Vanilla AT, and MART minimize the adversarial risk defined in Equation 5, in which the $\mathbb{1}(f_\theta(x_i') \neq y_i)$ is replaced by the CE loss under Vanilla AT and by the boosted cross-entropy (BCE) loss under MART. Formally, the boosted cross-entropy (BCE) loss is formulated as

$$BCE(p(x_i', \theta), y_i) = -\log p_{y_i}(x_i', \theta) - log(1 - \max_{k \neq y_i} p_k(x_i', \theta)); \tag{7}$$

which is built on the cross-entropy (CE) loss defined as

$$CE(p(x_i', \theta), y_i) = -\log p_{y_i}(x_i', \theta), \tag{8}$$

where $p_{y_i}(x_i', \theta)$ is the probability that the network predicts class $y_i$ given the input example $x_i'$. However, the CE loss only focuses on the probability that the input is assigned to its ground-truth class and does not place any constraint on the probability that the data point is assigned to a class other than its ground-truth class; hence, it does not specifically minimize the $Risk_{adv}$ (Equation 6). To motivate our choice for the proposed surrogate loss to be used in Equation 6, we consider a multi-class hinge loss developed for SVMs (Crammer & Singer, 2001) and the vector of class scores denoted by $f(x_i', \theta)$ is the logit output of the network, then $f(x_i', \theta) = (f_1(x_i', \theta), f_2(x_i', \theta), ......, f_C(x_i', \theta))$ and $s_j = f_j(\theta, x')$ represents the score of the j-th class. The multi-class SVM loss (hinge loss) for the i-th example is formalized as

$$l_i = \sum_{j \neq y_i} \max(0, s_j - s_{y_i} + \delta). \tag{9}$$

**Example**: Suppose there are three classes, and the vectors of classes' scores $s = [12, -6, 11]$; scores associated with "cat," "dog," and "ship," respectively. For illustration, let us assume the true class is "cat" (score is 12, i.e., $y_i = 0$). In addition, we assume our desired margin $\delta$ is 8.

Under our assumption, $l_i = \max(0, -6 - 12 + 8) + \max(0, 11 - 12 + 8) = 0 + 7$. Since the correct class score of 12 was greater than the incorrect class score of $-6$ by at least the margin of 8, we got zero loss on the first term. The second term $\max(0, 11 - 12 + 8) = 7$. Even though the correct class had a higher score than the incorrect class ($12 > 11$), it was not greater by the desired margin of 8. 7 represents how much higher the difference would have to be to meet the margin. This example illustrates the benefit of the margin loss in assessing the gap between the true class and other classes.

To penalize violated margins more strongly, we consider

$$l_i' = \sum_{j \neq y_i} \max(0, s_j - s_{y_i} + \delta)^2. \tag{10}$$

The illustrative example of the Multi-class SVM encourages the correct class's score to be higher than all other scores by at least a margin of $\delta$, imposing a margin gap between the true class and the other false classes' score. We can extend this formulation to a more complex setting. We exploit the multi-class classification hinge loss (margin-based loss) proposed for SVM (Crammer & Singer, 2001) to formulate a criterion that optimizes a multi-class classification hinge loss between the input $f_\theta(x_i')$ tensor and the output $y_i$. For each input, we minimize the loss:

$$L_i = \sum_{j \neq y_i} \max(0, (f_j(\theta, x') - f_{y_i}(\theta, x') + \delta)) \tag{11}$$

A robust classifier should correctly classify adversaries. For any input pair $(x_i, y_i)$, the corresponding adversarial pair $(x_i', y_i)$ should be classified correctly. We expect that if our classifier loss is minimized, then so is $\delta - f_{y_i}(\theta, x') + f_j(\theta, x')$ for $y_i \neq j$. This quantity is positive for all $y_i$ as long as the output of the classifier conditioned on the correct label is larger by at least $\delta$ than the classifier output conditioned on the rest of the labels. Therefore, we minimize $L_i$ to explicitly enforce this margin. Instead of focusing solely on the possibility of the model misclassifying the adversarial into the 2nd, 3rd, 4th, or fifth most probable false class, we consider a relaxed version that incorporates more classes (2nd, 3rd, 4th, 5th, up to the $qth$ most probable false classes where $q < |C|$). This relaxed version considers the first several most probable classes, making our adversarial risk formulation (Equation 6) less restrictive in terms of $y_k$. Under the relaxed version of the adversarial risk (Equation 6), (Li et al., 2021) minimizes $\sum_{j \neq y_i} \max(0, (P_j(\theta, x') - P_{y_i}(\theta, x') + \delta))$. However, based on our discussion on SVM loss, we consider the logits and penalize the violated margin strongly. Hence, to minimize the adversarial $Risk_{adv}$ (Equation 6), we minimize the loss

$$L_i' = \sum_{j \neq y_i} \max(0, (f_j(\theta, x') - f_{y_i}(\theta, x') + \delta))^2. \tag{12}$$

Equation 12 maximizes the adversarial probability gaps between the true and most probable false classes by applying a margin constraint, thus fulfilling the first condition outlined in the introduction. Conversely, our baseline losses rely on the cross-entropy loss, which prioritizes the probability that the input is assigned to its ground truth, thereby satisfying the second condition. Consequently, all the conditions (Conditions 1 and 2) enumerated in the introduction are met. It is important to note that the margin loss function significantly impacts a model's robustness to adversarial attacks. Increasing the margin encourages the model to create a greater separation between classes in the feature space. This larger separation helps make it more difficult for adversarial perturbations to push an input across the decision boundary and into a different class, thereby enhancing the model's robustness. Conversely, a smaller margin reduces the separation between classes, making the model more susceptible to adversarial attacks since less perturbation is needed to cause misclassification. While a more significant margin can boost robustness, it may also decrease natural accuracy if the margin becomes too large, as the model might prioritize robustness overfitting the training data effectively. Therefore, a careful selection of the margin parameter is crucial.

### 4.3 Exponential Moving Average (EMA) of logits

Various strategies have emerged to enhance model generalization, with one notable method being the weight averaging of model parameters (Polyak & Juditsky, 1992; Oord et al., 2018; Athiwaratkun et al., 2018; Izmailov et al., 2018). Recently, this approach has found application in GAN training (Yaz et al., 2018), and in bolstering adversarial robustness (Gowal et al., 2020). Our research introduces a novel weight-independent approach using logit averaging. We propose that reducing the discrepancy between the moving averages of natural and adversarial logits in the regularization term enhances adversarial robustness while maintaining reasonable natural accuracy. This approach minimizes the gap between natural and adversarial accuracy.

The process involves computing the moving average of logits $logit_t \leftarrow \tau * logit_{(t-1)} + (1-\tau) * logit$, where $logit$ denotes the current logit value, $logit_{(t-1)}$ represents the exponential moving average at previous stages, and $logit_t$ is the logit used in the regularization term. While (Atsague et al., 2023) minimize the disparity between natural and adversarial logits by employing the modified Huber (mHuber) model, which has demonstrated greater robustness to outliers and noisy data compared to the original Huber (Guo et al., 2021), we opt for the modified Huber loss to minimize the difference between the moving averages of natural and adversarial logits. By incorporating current and previous iteration logits through a moving average calculation, we gain valuable comparative insights from the early stages of training. The moving average integrates information from both natural and adversarial examples over time (shown in Appendix A), providing a more stable estimate of the model's predictions. Therefore, we minimize $mHuber(logit'_t, logit_t, \alpha)$ where $logit'_t$ and $logit_t$ represent the adversarial and natural moving averages of logits, respectively. we experimented on different values of $\tau$ and recorded our best performance when $\tau = 0.2$ (See Table 2 and 3).

### 4.4 Improvement Strategy

For illustration, we consider the vanilla AT(Madry et al., 2018) and TRADES (Zhang et al., 2019). The vanilla AT minimizes the cross-entropy (CE) loss defined by

$$CE(p(x'_i, \theta), y_i) = -\log p_{y_i}(x'_i, \theta);$$ (13)

In this scenario, adversarial examples used for training are generated using the CE losses. However, to enhance the vanilla AT, the CE loss is complemented with the margin-based loss. Consequently, the adversarial examples used for training are generated using the loss $L'_i + CE$ (inner maximization). For the outer minimization, we aim to minimize the loss

$$CE(p(x'_i, \theta), y_i) + L'_i + \beta * mHuber(logit'_t, logit_t, \alpha)$$ (14)

Where $logit'_t$ and $logit_t$ represent the adversarial and natural moving averages logit's, respectively. The improvement strategy adopted for the Vanilla AT can be expanded to other variants. For instance, TRADES minimize

$$CE(p(x_i, \theta), y_i) + \frac{1}{\lambda} . KL(p(x_i, \theta) || p(x'_i, \theta)).$$ (15)

To improve TRADES, we generate the adversarial examples using the loss $L'_i + CE$, and for training, we minimize the loss

$$L'_i + CE(p(x_i, \theta), y_i) + \frac{1}{\lambda} KL(p(x_i, \theta) || p(x'_i, \theta)) + \beta * mHuber(logit'_t, logit_t, \alpha).$$ (16)

Moreover, drawing inspiration from effective enhancement strategies proposed and implemented in previous studies, notably, the methodology detailed in PMHR-AT (Atsague et al., 2023), we introduce a streamlined yet remarkably effective training approach called Logits Moving Average Adversarial Training (LMA-AT), described in detail below.

$$L'_i + BCE(p(x'_i, \theta), y_i) + \beta * mHuber(logit'_t, logit_t, \alpha)$$ (17)

A notable difference between our proposed LMA-AT and existing methods, such as PMHR-AT, is that we regularize the adversarial loss by minimizing the disparity between the moving average of natural and adversarial logits. In contrast, PMHR-AT considered the logits, applied the $l_2$ penalty to the network weights, and reduced the gap between natural and adversarial accuracy by adjusting the strength of the regularization term based on the similarity between the predicted natural and adversarial class probability distributions. We do not use $l_2$ regularization on the network weights as this may be computationally intense or vary the regularization strength. Instead, we utilize the moving average of logits and the margin-based loss, resulting in better generalization and a reduced gap between natural and adversarial accuracy. We term the improved training objectives, Equation 14 and Equation 16, **Standard AT+Ours** and **TRADES+Ours** respectively. Similarly, in the following sections, **MART+Ours** refers to the improved version of MART. See Table 1 for additional details. Algorithm 1 illustrates the training strategy of **LMA-AT**. a similar approach is adopted under **Standard AT+Ours**, **TRADES+Ours** and **MART+Ours**.

Table 1: This table provides an overview of the enhanced versions of the baseline losses. The terms highlighted in bold represent the improvement strategies incorporated.

| Method | Improved Losses |
|---|---|
| Standard AT+**Ours** | $\boldsymbol{L_i'} + CE(p(x_i',\theta),y_i) + \beta * \boldsymbol{mHuber(logit_t',logit_t,\alpha)}$ |
| TRADES+**Ours** | $\boldsymbol{L_i'} + CE(p(x_i,\theta),y_i) + \frac{1}{\lambda}KL(p(x_i,\theta)||p(x_i',\theta)) + \beta * \boldsymbol{mHuber(logit_t',logit_t,\alpha)}$ |
| MART+**Ours** | $\boldsymbol{L_i'} + BCE(p(x_i',\theta),y_i) + \lambda \cdot KL(p(x_i,\theta)||p(x_i',\theta)) \cdot (1 - p_{y_i}(x_i,\theta)) + \beta * \boldsymbol{mHuber(logit_t',logit_t,\alpha)}$ |
| **LMA-AT(Ours)** | $\boldsymbol{L_i'} + BCE(p(x_i',\theta),y_i) + \beta * \boldsymbol{mHuber(logit_t',logit_t,\alpha)}$ |

---

**Algorithm 1** Training procedure of LMA-AT

---

**Input:** Training data $D = \{x_i, y_i\}_{i=1}^n$, step size $\mu_1$ and $\mu_2$ for the inner and the outer optimization respectively, the batch size $m$, the number of outer iteration $T$, the number of inner iteration $K$, the moving average parameter $\tau = 0.2$, $\alpha$, and the regularization parameter $\beta$.

**Initialization:**

Instantiate and initialize a model $f_\theta$

$logit_0 = 0$

$logit_0' = 0$

**for** $t = 1, 2, ...., T$ **do**

    At random, uniformly sample a mini-batch of training data $B_{(t)} = \{x_1, ..., x_m\}$

    **for** *each* $x_i \in B_{(t)}$ **do**

        $x_i' = x_i + 0.001 \times k; k \sim \mathcal{N}(\mathbf{0}, \mathbf{I})$

        **for** $k = 1, 2, ...., K$ **do**

            $x_i' = \prod_{B_\epsilon[x_i]}(x_i' + \mu_1 sgn(\nabla_{x_i'}[L_i' + CE(p(x_i',\theta),y_i)]))$

        **end**

    **end**

    $logit_t' \leftarrow \tau * logit_{t-1}' + (1 - \tau) * f(x_i', \theta)$

    $logit_t \leftarrow \tau * logit_{t-1} + (1 - \tau) * f(x_i, \theta)$

    $L_i' = \sum_{j \neq y_i} \max(0, (f_j(\theta, x') - f_{y_i}(\theta, x') + \delta))^2$

    $\theta = \theta - \frac{\mu_2}{m} \sum_{i=1}^m \nabla_\theta[L_i' + BCE(p(x_i',\theta),y_i) + \beta * mHuber(logit_t',logit_t,\alpha)]$

    $logit_{t-1}' = logit_t'$

    $logit_{t-1} = logit_t$

**end**

**Output:** $f_\theta$

---

In Algorithm 1, $L_i'(p(x_i',\theta),y_i)) = \sum_{j \neq y_i} \max(0, (f_j(\theta, x') - f_{y_i}(\theta, x') + \delta))^2$.

In summary, the Cross-entropy loss is a common choice in adversarial training (AT) for measuring the difference between predicted probabilities and true labels. In conventional AT, adversarial examples are often generated using techniques like PGD. While cross-entropy loss effectively maintains natural accuracy, its ability to improve robustness against adversarial attacks can be limited, particularly when facing highly optimized attacks. The Margin-Aware loss introduces a new approach by emphasizing confidence calibration and robustness. It enforces a margin between the logits of the correct class and those of incorrect classes, prompting the model to differentiate between correct and incorrect classifications more clearly. This mechanism helps push adversarial examples away from the decision boundary, improving robustness to subtle perturbations. A key innovation of our approach is the combination of Cross-Entropy Loss and Margin Loss, incorporating the moving average of logits as part of the regularization term. This method leverages the cross-entropy loss to maintain high natural accuracy while using the margin loss to bolster resilience against adversarial attacks. The margin size is a crucial hyperparameter chosen based on dataset complexity and the strength of the adversarial attacks. By integrating the moving average of logits into the regularization, we capture valuable information from the previous iteration, which helps to improve model robustness further. Our approach aims to balance natural accuracy and robust accuracy, unlike methods such as TRADES

or MART, which may significantly compromise clean accuracy for enhanced robustness. This balance is essential for real-world applications, where models must perform effectively on clean and adversarial inputs.

## 5 Experiments

We conducted a series of experiments and compared our method with the state-of-the-art defenses on benchmark datasets CIFAR-10 (Krizhevsky & Hinton, 2009), CIFAR-100 (Krizhevsky & Hinton, 2009), and TinyImageNet (Deng et al., 2009). We tested on two model architectures: ResNet-18 (He et al., 2016a) and a larger capacity network, WideResNet-34-10 (Zagoruyko & Komodakis, 2016a).
**Baselines**: We compare our approach with Vanilla AT (Madry et al., 2018) and the top-performing variants of adversarial training defenses to date: PMHR-AT(Atsague et al., 2023), TRADES (Zhang et al., 2019), and MART (Wang et al., 2020). Additionally, we benchmark our work against other margin-based approaches such as MMA (Ding et al., 2020), GAIRA (Zhang et al., 2020c), MAIL (Liu et al., 2021), and WAT (Zeng et al., 2021).

### 5.1 Training settings

The parameters are selected using the Ray Tune hyperparameter search tool proposed in (Liaw et al., 2018). For each model and dataset, we define the search range for $\beta$ as [1, 100] and for weight decay as (0, 0.2). The search range for $\tau$ is set to the interval [0, 1) and the one of $\delta$ is [0, 1]. For the parameter $\alpha$, we base the search range on the recommended value of $\alpha = 1.345$ from the original Huber loss suggested by (Bach et al., 2011; Guo et al., 2021). We increment this value by 1, resulting in a search range of 1.345, 2.345, 3.345, 4.345, ..., up to 9.345. Our search results identified the following best parameters.

Under ResNet-18, $\alpha$ is 6.345 on TinyImageNet, 5.345 on CIFAR-10, and CIFAR-100. On WRN-34-10, $\alpha$ is 2.345. For TRADES, $\frac{1}{\lambda}$ is set to 6.0, and $\lambda$ is 5.0 in MART as specified in their original papers. We consider the same parameters defined in their original papers for other baselines. All the models are trained using SGD for 130 epochs with momentum 0.9 and the batch size m=100. The initial learning rate is 0.01, then decayed by a factor of ten at the 75th and further decayed at the 90th epoch. We consider the weight decay of 3.5e-3. Adversarial data used in training are generated using PGD with a random start, maximum perturbation $\epsilon$ set to 8/255, step size as 2/255, and the number of steps is 10. Our best performances are recorded when the margin $\delta$ is set to 0.9, The regularization parameter $\beta$ is set to 96 on TinyImageNet and CIFAR-100, 86 on CIFAR-10.

### 5.2 Evaluation details

We evaluated our method under **White-box attack** threats including the $L_\infty$ PGD-20/100 (Madry et al., 2018), FGSM (Goodfellow et al., 2014), CW (PGD optimized with CW loss, confidence level $K$=50) (Carlini & Wagner, 2017), and on **Ensemble of Attacks** such as AutoAttack (Croce & Hein, 2020b), which consisting of APGD-CE (Croce & Hein, 2020b), APGD-T (Croce & Hein, 2020b), FAB-T (Croce & Hein, 2020a), and Square (a black-box attack). Under **White-box attack**, The perturbation size is set to $\epsilon$=8/255, and the step size is 1/255. Additionally, we evaluated on strong **Black-box** attacks SQUARE (Andriushchenko et al., 2020) and SPSA (Uesato et al., 2018) with the perturbation size of 0.001 (for gradient estimation), sample size of 100, 20 iterations, and learning rate 0.01.

### 5.3 Experimental results

#### 5.3.1 Sensitivity to moving average Hyperparameter

We conducted a series of experiments to assess the effectiveness of using the moving average of logits to improve model performance. In this experiment, we consider our proposed loss: Logits Moving Average Adversarial Training (LMA-AT). By varying the moving average parameter $0 \leq \tau < 1$, we adjusted the contribution of the moving average throughout the training process. This process involves computing the moving average of logits, $logit_t \leftarrow \tau logit_{(t-1)} + (1 - \tau) * logit$, where $logit$ denotes the current logit value,

$logit_{(t-1)}$ represents the exponential moving average from previous stages, and $logit_t$ is the logit used in the regularization term. Increasing $\tau$ increases the influence of the moving average on the overall performance.

Table 2: Assessing performance across various values of our moving average parameter, $\tau$, under CIFAR-10 with ResNet18 architecture.

| $\tau$ | Natural | PGD-20 | PGD-100 | CW | SPSA | AA |
|---|---|---|---|---|---|---|
| 0.0 | $79.33_{\pm 0.001}$ | $56.93_{\pm 0.003}$ | $55.92_{\pm 0.001}$ | $51.97_{\pm 0.004}$ | $58.86_{\pm 0.002}$ | $48.34_{\pm 0.005}$ |
| 0.1 | $84.11_{\pm 0.007}$ | $56.45_{\pm 0.001}$ | $54.65_{\pm 0.002}$ | $52.62_{\pm 0.030}$ | $60.14_{\pm 0.007}$ | $48.75_{\pm 0.001}$ |
| **0.2** | $83.56_{\pm 0.0021}$ | $57.21_{\pm 0.001}$ | $55.64_{\pm 0.0012}$ | $52.30_{\pm 0.001}$ | $60.44_{\pm 0.001}$ | $49.10_{\pm 0.001}$ |
| 0.7 | $81.72_{\pm 0.001}$ | $57.83_{\pm 0.004}$ | $56.47_{\pm 0.003}$ | $52.49_{\pm 0.031}$ | $59.10_{\pm 0.001}$ | $48.96_{\pm 0.003}$ |
| 0.9 | $80.33_{\pm 0.001}$ | $57.31_{\pm 0.006}$ | $56.12_{\pm 0.001}$ | $52.28_{\pm 0.001}$ | $59.16_{\pm 0.003}$ | $49.20_{\pm 0.005}$ |

Table 3: Assessing performance across various values of our moving average parameter, $\tau$, under CIFAR-100 with ResNet18 architecture.

| $\tau$ | Natural | PGD-20 | PGD-100 | CW | SPSA | AA |
|---|---|---|---|---|---|---|
| 0.0 | $51.40_{\pm 0.002}$ | $32.76_{\pm 0.006}$ | $32.21_{\pm 0.001}$ | $28.43_{\pm 0.008}$ | $31.71_{\pm 0.002}$ | $26.51_{\pm 0.003}$ |
| 0.1 | $59.78_{\pm 0.004}$ | $32.02_{\pm 0.001}$ | $31.08_{\pm 0.005}$ | $28.91_{\pm 0.006}$ | $34.98_{\pm 0.001}$ | $25.51_{\pm 0.003}$ |
| **0.2** | $58.86_{\pm 0.013}$ | $32.51_{\pm 0.02}$ | $31.65_{\pm 0.041}$ | $29.04_{\pm 0.021}$ | $34.07_{\pm 0.032}$ | $26.29_{\pm 0.011}$ |
| 0.7 | $53.98_{\pm 0.008}$ | $33.48_{\pm 0.003}$ | $32.83_{\pm 0.006}$ | $29.31_{\pm 0.003}$ | $32.78_{\pm 0.001}$ | $26.84_{\pm 0.002}$ |
| 0.9 | $52.20_{\pm 0.001}$ | $33.29_{\pm 0.006}$ | $32.83_{\pm 0.001}$ | $29.19_{\pm 0.002}$ | $31.88_{\pm 0.001}$ | $26.78_{\pm 0.005}$ |

In Table 2 and Table 3, we experimented with different values of $\tau$ and highlighted the $\tau$ values that yielded our overall best performance in bold. The overall best performance is recorded for $\tau = 0.2$ (This best parameter is recorded using Ray Tune as described in Section 5.1).

The results presented in 2 and 3 demonstrate that significant changes occur when the moving average parameter $\tau$ is varied. For example, in both tables, comparing the results with $\tau = 0.1$ to those with $\tau = 0.9$ reveals a consistent improvement in robust accuracy under PGD-20/100 and AutoAttack, accompanied by a notable drop in natural accuracy. A higher $\alpha$ (closer to 1) places greater emphasis on recent logits, potentially leading to less historical data retention. In this scenario, information gain primarily concentrates on recent logits, which may overlook long-term patterns. Conversely, a lower $\alpha$ (closer to 0) assigns more weight to past logits, depending on how much of the previous information we want to consider, that can significantly impact the model's robustness and lead to poor trade-off between natural and robust accuracy. The choice of $\alpha$ represents a trade-off between adapting to new data and preserving historical information. Identifying the optimal $\alpha$ is essential, as it determines the extent of information retained over time and the model's ability to adapt to changing patterns. Varying the moving average parameter $\alpha$ influences information gain by adjusting the emphasis on recent versus past data. This balance is crucial, as it impacts both natural and adversarial accuracy, making the selection of the best $\alpha$ value vital to achieving adversarial robustness and batter trade-off between natural and adversarial accuracy.

### 5.3.2 Sensitivity to the regularization Hyperparameter

To evaluate the effect of the regularization hyperparameter $\beta$ on our proposed loss function (LMA-AT), we use the training and evaluation setups described in Sections 5.1 and 5.2. We experiment with various values of the regularization parameter, and the results are presented in Tables 4 and 5.

Table 4: Assessing performance across various values of the parameter, $\beta$, under **CIFAR-10 with ResNet18** architecture.

| $\beta$ | Natural | PGD-20 | PGD-100 | CW | AA |
|---|---|---|---|---|---|
| 80 | $84.27_{\pm0.011}$ | $55.92_{\pm0.001}$ | $54.67_{\pm0.003}$ | $52.08_{\pm0.002}$ | $47.70_{\pm0.032}$ |
| 82 | $83.33_{\pm0.01}$ | $56.73_{\pm0.023}$ | $55.08_{\pm0.011}$ | $52.48_{\pm0.006}$ | $48.68_{\pm0.014}$ |
| 84 | $83.68_{\pm0.012}$ | $56.54_{\pm0.012}$ | $54.82_{\pm0.005}$ | $52.37_{\pm0.002}$ | $48.60_{\pm0.061}$ |
| 86 | $83.56_{\pm0.021}$ | $57.21_{\pm0.001}$ | $55.64_{\pm0.012}$ | $52.30_{\pm0.001}$ | $49.10_{\pm0.001}$ |
| 88 | $83.50_{\pm0.031}$ | $56.90_{\pm0.022}$ | $55.21_{\pm0.013}$ | $52.42_{\pm0.003}$ | $48.71_{\pm0.051}$ |
| 90 | $83.28_{\pm0.014}$ | $56.87_{\pm0.031}$ | $55.12_{\pm0.034}$ | $52.29_{\pm0.002}$ | $48.66_{\pm0.012}$ |
| 92 | $83.43_{\pm0.022}$ | $57.11_{\pm0.021}$ | $55.21_{\pm0.013}$ | $52.39_{\pm0.002}$ | $48.51_{\pm0.031}$ |

Table 5: Assessing performance across various values of the parameter, $\beta$, under **CIFAR-100 with ResNet18** architecture.

| $\beta$ | Natural | PGD-20 | PGD-100 | CW | AA |
|---|---|---|---|---|---|
| 80 | $59.47_{\pm0.012}$ | $31.92_{\pm0.021}$ | $30.77_{\pm0.005}$ | $28.71_{\pm0.021}$ | $25.38_{\pm0.035}$ |
| 82 | $59.02_{\pm0.033}$ | $32.29_{\pm0.021}$ | $31.31_{\pm0.004}$ | $28.69_{\pm0.012}$ | $25.48_{\pm0.031}$ |
| 84 | $58.99_{\pm0.012}$ | $32.58_{\pm0.011}$ | $31.55_{\pm0.013}$ | $29.02_{\pm0.024}$ | $26.16_{\pm0.041}$ |
| 86 | $58.86_{\pm0.013}$ | $32.51_{\pm0.002}$ | $31.65_{\pm0.041}$ | $29.04_{\pm0.021}$ | $26.29_{\pm0.011}$ |
| 88 | $58.73_{\pm0.022}$ | $32.38_{\pm0.023}$ | $31.52_{\pm0.026}$ | $28.95_{\pm0.013}$ | $26.07_{\pm0.035}$ |
| 90 | $58.92_{\pm0.009}$ | $32.09_{\pm0.021}$ | $31.20_{\pm0.003}$ | $28.7a_{\pm0.002}$ | $25.77_{\pm0.032}$ |
| 92 | $59.25_{\pm0.011}$ | $31.85_{\pm0.001}$ | $31.20_{\pm0.003}$ | $28.54_{\pm0.002}$ | $25.45_{\pm0.032}$ |

Let's discuss the impact of the hyperparameter $\beta$ on model robustness. As expected, increasing the regularization parameter leads to improved robust accuracy. For example, using ResNet18 on CIFAR-10, we observed increases in robust accuracy of 1.29%, 0.97%, and 1.4% under PGD-20, PGD-100, and AA, respectively, when $\beta$ was increased from 80 to 86. These improvements were achieved with only a minor drop in natural accuracy. Our optimal parameter setting ($\beta = 86$) provided a better trade-off between natural accuracy and adversarial robustness than existing AT-based defense methods. A similar pattern was observed on CIFAR-100, where robust accuracy increased by 0.59%, 0.88%, 0.33%, and 0.78% under PGD-20, PGD-100, CW, and AA, respectively. The values of these hyperparameters were heuristically determined based on our optimal parameters. Additional experiments on TinyImageNet are reported in **Appendix B**. Since the regularization parameter $\beta$ controls the impact of the regularization term on the training process, a larger $\beta$ places more emphasis on minimizing the disparity between the moving averages of natural and adversarial logits, thereby reducing the gap between them. This encourages the model to produce similar predictions for both types of inputs, ensuring that adversarial examples have less impact on the final prediction. As a result, the model becomes less sensitive to small perturbations, reducing the effectiveness of adversarial attacks and enhancing adversarial robustness. Additionally, minimizing the difference between the logits acts as a regularizer, promoting smoothness in the model's decision boundary. A smoother decision boundary implies that the model is less likely to be swayed by adversarial examples close to natural examples in the input space.

### 5.3.3 Effectiveness of our proposed method

Table 6 presents the results for CIFAR-10 using the ResNet-18 model. Tables 7 and 10 show the results for CIFAR-10 using the WideResNet-34-10 model. Additionally, we evaluated the ResNet-18 model on CIFAR-100 and TinyImageNet datasets, with the results reported in Tables 8 and 9, respectively.

Table 6: Clean and robust accuracy on **ResNet-18** and Under **CIFAR-10**. We perform six runs and report the average performance with 95% confidence intervals. The 'Clean' column represents accuracy on natural examples.

| *Method* | Clean | FGSM | PGD-20 | PGD-100 | CW | AA | SQUARE | SPSA |
|---|---|---|---|---|---|---|---|---|
| vanillaAT | $\mathbf{85.80}_{\pm 0.001}$ | $57.87_{\pm 0.0023}$ | $52.05_{\pm 0.003}$ | $49.28_{\pm 0.0022}$ | $51.08_{\pm 0.001}$ | $46.62_{\pm 0.004}$ | $55.69_{\pm 0.0014}$ | $56.17_{\pm 0.001}$ |
| TRADES | $82.46_{\pm 0.0012}$ | $58.26_{\pm 0.0030}$ | $54.78_{\pm 0.0010}$ | $53.45_{\pm 0.0032}$ | $51.65_{\pm 0.0021}$ | $49.08_{\pm 0.0031}$ | $55.64_{\pm 0.0011}$ | $56.50_{\pm 0.0020}$ |
| MART | $81.30_{\pm 0.003}$ | $58.06_{\pm 0.001}$ | $54.73_{\pm 0.006}$ | $53.28_{\pm 0.005}$ | $51.86_{\pm 0.0031}$ | $49.01_{\pm 0.0020}$ | $55.66_{\pm 0.0031}$ | $56.15_{\pm 0.0040}$ |
| PMHR−AT | $83.12_{\pm 0.0022}$ | $60.34_{\pm 0.0010}$ | $56.13_{\pm 0.0021}$ | $54.45_{\pm 0.0031}$ | $52.16_{\pm 0.0010}$ | $49.42_{\pm 0.0020}$ | $56.54_{\pm 0.00021}$ | $57.16_{\pm 0.0003}$ |
| **vanillaAT + Ours** | $82.82_{\pm 0.001}$ | $59.89_{\pm 0.0013}$ | $56.36_{\pm 0.002}$ | $54.83_{\pm 0.0021}$ | $51.95_{\pm 0.004}$ | $48.32_{\pm 0.002}$ | $57.11_{\pm 0.01}$ | $60.35_{\pm 0.003}$ |
| **TRADES + Ours** | $83.93_{\pm 0.0012}$ | $59.32_{\pm 0.0007}$ | $56.23_{\pm 0.0021}$ | $54.98_{\pm 0.0011}$ | $51.73_{\pm 0.0024}$ | $48.78_{\pm 0.0021}$ | $58.46_{\pm 0.0012}$ | $59.45_{\pm 0.0020}$ |
| **MART + Ours** | $83.33_{\pm 0.012}$ | $60.87_{\pm 0.003}$ | $\mathbf{57.41}_{\pm 0.003}$ | $\mathbf{55.81}_{\pm 0.006}$ | $51.83_{\pm 0.0041}$ | $48.48_{\pm 0.0013}$ | $58.54_{\pm 0.0042}$ | $\mathbf{60.56}_{\pm 0.0025}$ |
| **LMA−AT(Ours)** | $83.56_{\pm 0.0021}$ | $\mathbf{61.19}_{\pm 0.001}$ | $57.21_{\pm 0.001}$ | $55.64_{\pm 0.012}$ | $\mathbf{52.30}_{\pm 0.001}$ | $49.10_{\pm 0.001}$ | $\mathbf{59.54}_{\pm 0.001}$ | $60.44_{\pm 0.001}$ |

Table 7: Clean and robust accuracies on **WRN-34-10** and Under **CIFAR-10**. We perform six runs and report the average performance with 95% confidence intervals. The 'Clean' column represents accuracy on natural examples.

| *Method* | Clean | FGSM | PGD-20 | PGD-100 | CW | AA | SQUARE | SPSA |
|---|---|---|---|---|---|---|---|---|
| vanillaAT | $\mathbf{86.46}_{\pm 0.0013}$ | $61.62_{\pm 0.0021}$ | $56.75_{\pm 0.002}$ | $54.72_{\pm 0.001}$ | $55.63_{\pm 0.0012}$ | $51.06_{\pm 0.0023}$ | $59.68_{\pm 0.0012}$ | $60.66_{\pm 0.002}$ |
| TRADES | $84.58_{\pm 0.0021}$ | $60.60_{\pm 0.001}$ | $57.71_{\pm 0.0012}$ | $56.69_{\pm 0.002}$ | $55.01_{\pm 0.0013}$ | $52.57_{\pm 0.002}$ | $59.45_{\pm 0.0024}$ | $61.09_{\pm 0.0023}$ |
| MART | $84.25_{\pm 0.001}$ | $62.03_{\pm 0.00}$ | $58.29_{\pm 0.0032}$ | $55.56_{\pm 0.0011}$ | $54.82_{\pm 0.00}$ | $51.40_{\pm 0.00}$ | $58.21_{\pm 0.00}$ | $59.87_{\pm 0.00}$ |
| PMHR−AT | $84.87_{\pm 0.0020}$ | $63.05_{\pm 0.0010}$ | $59.26_{\pm 0.0021}$ | $57.60_{\pm 0.0031}$ | $\mathbf{56.36}_{\pm 0.0010}$ | $\mathbf{53.58}_{\pm 0.002}$ | $59.67_{\pm 0.0021}$ | $61.18_{\pm 0.001}$ |
| **vanillaAT + Ours** | $85.96_{\pm 0.002}$ | $63.03_{\pm 0.0013}$ | $59.76_{\pm 0.005}$ | $58.31_{\pm 0.0011}$ | $56.03_{\pm 0.002}$ | $52.82_{\pm 0.001}$ | $60.02_{\pm 0.0014}$ | $63.78_{\pm 0.003}$ |
| **TRADES + Ours** | $85.62_{\pm 0.0032}$ | $63.22_{\pm 0.007}$ | $59.31_{\pm 0.021}$ | $58.26_{\pm 0.0011}$ | $54.87_{\pm 0.024}$ | $52.25_{\pm 0.0021}$ | $58.89_{\pm 0.0012}$ | $63.95_{\pm 0.0020}$ |
| **MART + Ours** | $84.83_{\pm 0.004}$ | $63.66_{\pm 0.003}$ | $\mathbf{60.89}_{\pm 0.005}$ | $\mathbf{59.76}_{\pm 0.001}$ | $55.56_{\pm 0.0021}$ | $52.45_{\pm 0.003}$ | $59.42_{\pm 0.0022}$ | $62.65_{\pm 0.002}$ |
| **LMA−AT(Ours)** | $85.39_{\pm 0.002}$ | $\mathbf{64.04}_{\pm 0.0012}$ | $60.62_{\pm 0.001}$ | $59.48_{\pm 0.0021}$ | $56.07_{\pm 0.001}$ | $52.61_{\pm 0.0024}$ | $\mathbf{60.10}_{\pm 0.005}$ | $\mathbf{64.19}_{\pm 0.001}$ |

The results of Table 6 and Table 7 demonstrate that our proposed method significantly improves the vanilla AT, TRADES, and MART. For instance, under ResNet-18 and WRN-34-10, respectively, the Vanilla AT improved by 2% and 3% on PGD-20, 5.55% and 3.59% on PGD-100, 0.87% and 0.5% on CW, 1.7% and 1.76% on AA, 1.42% and 0.34% on SQUARE, and 4.18% and 3.12% on SPSA. MART improves by 2.03% on clean accuracy under ResNet-18. Under ResNet-18 and WRN-34-10, respectively, MART improved by 2.68% and 2.6% on PGD-20, 2.53% and 4.2% on PGD-100, 2.88% and 1.21% on SQUARE, and 4.41% and 2.78% on SPSA. In addition, on AA, MART improves by 1.05% on AA under WRN-34-10. On the other hand, the improvement of TRADES is more visible on ResNet-18 with a 1.47% increase in Clean accuracy, 1.06% on FGSM, 1.45% on PGD-20, 1.53% on PGD-100, 2.82% on SQUARE, and 2.95% on SPSA. On WRN-34-10, TRADES improves by 1.04% on Clean accuracy, 2.62% on FGSM, 1.6% on PGD-20, 1.57% on PGD-100 and 2.86% SPSA. The overall best performance is recorded under LMA-AT.

The robustness of the proposed 'TRADES + Ours' and 'MART + Ours' methods is slightly lower under AutoAttack than TRADES and MART. However, a closer examination reveals that TRADES and MART achieve this robustness at the expense of natural accuracy. In contrast, 'TRADES + Ours' and 'MART + Ours' demonstrate better robust accuracy under FGSM, PGD-20, PGD-100, and stronger Black Box attacks such as SQUARE and SPSA. While PMHR-AT exhibits strong robustness under AutoAttack, it does so at the cost of natural accuracy. Our method outperforms PMHR-AT in terms of natural accuracy, as well as robustness under FGSM, PGD-20/100, and stronger Black Box attacks like SQUARE and SPSA. Overall, our proposed method improves adversarial robustness in certain attacks and offers a better trade-off between natural and adversarial accuracy than existing approaches.

Table 8: Clean and robust accuracies on **ResNet-18** and Under **CIFAR-100**. We perform six runs and report the average performance with 95% confidence intervals. The 'Clean' column represents accuracy on natural examples.

| Method | Clean | FGSM | PGD-20 | PGD-100 | CW | AA | SQUARE | SPSA |
|---|---|---|---|---|---|---|---|---|
| vanillaAT | $56.87_{\pm0.0031}$ | $31.21_{\pm0.021}$ | $29.33_{\pm0.010}$ | $28.46_{\pm0.010}$ | $26.33_{\pm0.030}$ | $23.69_{\pm0.012}$ | $30.06_{\pm0.030}$ | $31.63_{\pm0.040}$ |
| TRADES | $57.16_{\pm0.0010}$ | $31.45_{\pm0.021}$ | $30.32_{\pm0.021}$ | $29.48_{\pm0.021}$ | $25.16_{\pm0.031}$ | $25.18_{\pm0.031}$ | $30.46_{\pm0.022}$ | $32.06_{\pm0.014}$ |
| MART | $54.02_{\pm0.0013}$ | $32.81_{\pm0.020}$ | $31.13_{\pm0.014}$ | $30.14_{\pm0.011}$ | $26.98_{\pm0.010}$ | $24.83_{\pm0.012}$ | $31.17_{\pm0.016}$ | $32.45_{\pm0.014}$ |
| PMHR−AT | $57.55_{\pm0.021}$ | $34.33_{\pm0.0031}$ | $32.25_{\pm0.021}$ | $31.35_{\pm0.014}$ | $27.78_{\pm0.011}$ | $25.96_{\pm0.031}$ | $31.32_{\pm0.015}$ | $32.60_{\pm0.04}$ |
| **vanillaAT** + **Ours** | $60.41_{\pm0.06}$ | $33.61_{\pm0.013}$ | $30.83_{\pm0.051}$ | $29.65_{\pm0.011}$ | $28.89_{\pm0.022}$ | $25.14_{\pm0.051}$ | $33.10_{\pm0.014}$ | $\mathbf{34.42}_{\pm0.031}$ |
| **TRADES** + **Ours** | $\mathbf{59.23}_{\pm0.012}$ | $34.05_{\pm0.08}$ | $31.72_{\pm0.021}$ | $30.98_{\pm0.011}$ | $27.84_{\pm0.023}$ | $25.42_{\pm0.021}$ | $31.66_{\pm0.012}$ | $33.30_{\pm0.063}$ |
| **MART** + **Ours** | $55.55_{\pm0.024}$ | $34.74_{\pm0.051}$ | $\mathbf{32.92}_{\pm0.033}$ | $\mathbf{32.29}_{\pm0.011}$ | $28.70_{\pm0.021}$ | $26.29_{\pm0.010}$ | $31.49_{\pm0.0345}$ | $33.57_{\pm0.032}$ |
| **LMA−AT(Ours)** | $58.86_{\pm0.013}$ | $\mathbf{34.79}_{\pm0.052}$ | $32.51_{\pm0.02}$ | $31.65_{\pm0.041}$ | $\mathbf{29.04}_{\pm0.021}$ | $\mathbf{26.29}_{\pm0.011}$ | $\mathbf{33.57}_{\pm0.016}$ | $34.07_{\pm0.032}$ |

The results in Tables 6, 7, and 8 demonstrate a consistent improvement in robust accuracy across different models and datasets when compared to baselines such as vanilla AT, TRADES, MART, and PMHR-AT. Although vanilla AT attains higher clean accuracy, it does so at the cost of significantly lower adversarial accuracy. The results demonstrate that our method not only outperforms other adversarial training (AT) variants in clean accuracy but also surpasses existing methods in most attack scenarios. Our proposed loss function consistently outperforms these baselines under various attacks, including FGSM, PGD-20, PGD-100, SQUARE, and SPSA. Additionally, despite the increase in robust accuracy, the reduction in natural accuracy remains minimal, indicating that the trade-off between natural and adversarial accuracy is effectively managed. Notably, on the more challenging CIFAR-100 dataset (as reported in Table 8), our method surpasses all baselines in both natural and adversarial accuracy, maintaining a better balance between robustness and accuracy, highlighting the superior performance of our proposed method, LMA-AT.

Table 9: Clean and robust accuracies on **TinyImageNet**, **ResNet-18** . We perform six runs and report the average performance with 95% confidence intervals. The 'Clean' column represents accuracy on natural examples.

| Method | Clean | PGD-20 | CW | AA |
|---|---|---|---|---|
| TRADES | $49.56_{\pm0.001}$ | $22.90_{\pm0.0021}$ | $19.70_{\pm0.0011}$ | $16.78_{\pm0.001}$ |
| MART | $45.94_{\pm0.003}$ | $26.02_{\pm0.002}$ | $21.87_{\pm0.001}$ | $19.20_{\pm0.002}$ |
| **TRADES** + **Ours** | $\mathbf{50.43}_{\pm0.0012}$ | $24.82_{\pm0.0021}$ | $20.52_{\pm0.0020}$ | $\mathbf{18.15}_{\pm0.0021}$ |
| **MART** + **Ours** | $46.88_{\pm0.002}$ | $\mathbf{26.87}_{\pm0.003}$ | $22.10_{\pm0.0021}$ | $\mathbf{19.84}_{\pm0.001}$ |
| **LMA−AT(Ours)** | $49.10_{\pm0.001}$ | $26.35_{\pm0.003}$ | $\mathbf{22.40}_{\pm0.006}$ | $18.31_{\pm0.001}$ |

For a more challenging task of classifying TinyImageNet, as presented in Table 9, our method outperforms both TRADES and MART under PGD-20 and CW attacks while maintaining a natural accuracy comparable to TRADES. However, TRADES exhibits lower robust accuracy compared to our method. Although MART achieves decent robust accuracy on PGD-20 and AA, it comes at the expense of a significant drop in natural accuracy. In contrast, our method achieves a better balance between robustness and accuracy.

Table 10: Clean and robust accuracies of different margin-based methods on **CIFAR-10** using the **WRN-34-10** model. Results are based on six runs, with the average performance reported along with 95% confidence intervals. The 'Clean' column indicates the accuracy of unperturbed examples.

| $Method$ | Clean | PGD-20 | CW | AA | SPSA |
|---|---|---|---|---|---|
| MMA | $86.21_{\pm 0.003}$ | $57.17_{\pm 0.0021}$ | $\mathbf{57.52}_{\pm 0.011}$ | $44.57_{\pm 0.0011}$ | $59.87_{\pm 0.011}$ |
| WAT | $85.16_{\pm 0.003}$ | $56.69_{\pm 0.002}$ | $54.06_{\pm 0.014}$ | $49.87_{\pm 0.021}$ | $60.78_{\pm 0.002}$ |
| MAIL | $\mathbf{86.82}_{\pm 0.003}$ | $60.38_{\pm 0.012}$ | $51.48_{\pm 0.001}$ | $47.15_{\pm 0.001}$ | $59.23_{\pm 0.032}$ |
| GAIRAT | $85.39_{\pm 0.005}$ | $60.59_{\pm 0.016}$ | $45.08_{\pm 0.014}$ | $42.30_{\pm 0.007}$ | $52.32_{\pm 0.004}$ |
| **vanillaAT + Ours** | $85.96_{\pm 0.002}$ | $59.76_{\pm 0.005}$ | $56.03_{\pm 0.002}$ | $\mathbf{52.82}_{\pm 0.001}$ | $63.78_{\pm 0.003}$ |
| **TRADES + Ours** | $85.62_{\pm 0.0032}$ | $59.31_{\pm 0.0021}$ | $54.87_{\pm 0.0024}$ | $52.25_{\pm 0.0021}$ | $63.95_{\pm 0.0020}$ |
| **MART + Ours** | $84.83_{\pm 0.0021}$ | $\mathbf{60.89}_{\pm 0.005}$ | $55.56_{\pm 0.0021}$ | $52.45_{\pm 0.003}$ | $62.65_{\pm 0.002}$ |
| **LMA−AT(Ours)** | $85.39_{\pm 0.002}$ | $60.62_{\pm 0.001}$ | $56.07_{\pm 0.001}$ | $52.61_{\pm 0.0024}$ | $\mathbf{64.19}_{\pm 0.001}$ |

The results of 8 and 9 show that our proposed LMA-AT method significantly outperforms the vanilla AT, TRADES, MART, and PMHR-AT. On CIFAR-100, TRADES + Ours improve TRADES by 2.07% on Clean accuracy, 2.6% on FGSM, 1.4% on PGD-20, 1.5% on PGD-100, 2.68% on CW, 1.2% on SQUARE, and 1.24% on SPSA. On the other hand, MART + Ours improve MART by 1.53% on clean accuracy, 1.93% on FGSM, 1.79% on PGD-20, 2.15% on PGD-100, 1.72% on CW, 1.46% on AA, and 1.12% on SPSA. Furthermore, our method was evaluated on TinyImageNet, where Table 9 illustrates substantial enhancements over TRADES and MART in Clean accuracy, PGD-20, CW, and AA metrics. Our LMA-AT method, demonstrating its efficacy, achieves a minimal gap between natural and adversarial accuracy. Additionally, our comparison with other margin-based approaches, detailed in Table 10, reveals that LMA-AT strikes a better balance between natural accuracy and adversarial robustness than these existing methods. Notably, our method outperforms other margins-based defenses by significant margins, such as GAIRAT by 10.31%, MAIL by 5.46%, WAT by 2.74%, and MMA by 8.04% on AA.

## 6 Ablation Studies

First, to evaluate the impact of the moving average of logits on overall adversarial robustness, we consider our proposed loss: Logits Moving Average Adversarial Training (LMA-AT). We varied the moving average parameter $\tau$ and recorded the results in Tables 2 and 3, where $\tau = 0.0$ represents no moving average applied. Both tables show poor performance under this condition. For instance, in Table 2, compared to the performance without the moving average, when $\tau = 0.2$, the improvement gap is 4.23% in natural accuracy, 0.28% in PGD-20, 0.33% in CW, 1.58% in SPSA, and 0.76% in AA. In Table 3, applying the moving average of logits ($\tau = 0.2$) resulted in an accuracy increase of 7.46% under natural accuracy, 0.61% on CW, and 2.36% on SPSA, while maintaining comparable performance against other attacks. Such an increase confirms the contribution of the moving average of logits to the overall robustness, providing a better trade-off between natural and adversarial accuracy. Lastly, we investigated the impact of the margin-based loss on two key aspects: the generation of adversarial samples used for training and the loss function applied during training. The options are summarized in the following table.

Table 11: This table offers an overview of different training settings, enabling the assessment of margin loss during both training and the generation of adversarial examples used in training.

| Options | Adversarial Loss | Training Loss |
|---|---|---|
| **A** | CE | $\mathbf{L'_i} + CE(p(x'_i, \theta), y_i) + \beta * \mathbf{mHuber(logit'_t, logit_t, \alpha)}$ |
| **B** | $\mathbf{L'_i} + CE$ | $BCE(p(x'_i, \theta), y_i) + \beta * \mathbf{mHuber(logit'_t, logit_t, \alpha)}$ |
| **C** | $\mathbf{L'_i} + CE$ | $\mathbf{L'_i} + BCE(p(x'_i, \theta), y_i) + \beta * \mathbf{mHuber(logit'_t, logit_t, \alpha)}$ |
| **D** | $\mathbf{L'_i} + CE$ | $\mathbf{L'_i} + BCE(p(x'_i, \theta), y_i)$ |

Under option **A**, the cross-entropy loss (CE) is used to generate the adversarial samples for training, and the margin loss is incorporated into the loss function used to train the model. In contrast, under option **B**, the

cross-entropy loss is supplemented with the margin-based loss for generating the adversarial samples used for training, but the margin loss is not included in the training loss function. Under option **C**, the margin loss contributes to both the adversarial data generation and the training processes. Finally, in Option D, we combined the cross-entropy loss with the margin loss to generate the adversarial examples used for training (inner maximization). The training loss is similar to option C. Still, we excluded the mHube regularization term, meaning no moving average was applied, allowing us to directly assess the impact of the mHuber regularization on the moving average of logits and its effect on model robustness.

Table 12: Clean and robust accuracy on **ResNet-18** and Under **CIFAR-10**. We perform six runs and report the average performance with 95% confidence intervals. The 'Clean' column represents accuracy on natural examples.

| *Method* | Clean | FGSM | PGD-20 | PGD-100 | CW | AA | SQUARE | SPSA |
|---|---|---|---|---|---|---|---|---|
| **A** | $83.88_{\pm 0.003}$ | $60.78_{\pm 0.001}$ | $56.26_{\pm 0.006}$ | $54.79_{\pm 0.005}$ | $51.91_{\pm 0.0031}$ | $48.57_{\pm 0.0020}$ | $56.83_{\pm 0.0031}$ | $60.55_{\pm 0.004}$ |
| **B** | $82.60_{\pm 0.001}$ | $60.36_{\pm 0.0013}$ | $56.74_{\pm 0.002}$ | $55.08_{\pm 0.0021}$ | $52.17_{\pm 0.004}$ | $48.67_{\pm 0.002}$ | $57.69_{\pm 0.001}$ | $60.51_{\pm 0.003}$ |
| **C** | $83.56_{\pm 0.0021}$ | $\mathbf{61.19}_{\pm 0.001}$ | $\mathbf{57.21}_{\pm 0.001}$ | $55.64_{\pm 0.012}$ | $\mathbf{52.30}_{\pm 0.001}$ | $\mathbf{49.10}_{\pm 0.001}$ | $\mathbf{59.54}_{\pm 0.001}$ | $\mathbf{60.44}_{\pm 0.001}$ |
| **D** | $\mathbf{84.42}_{\pm 0.0024}$ | $60.65_{\pm 0.004}$ | $55.77_{\pm 0.012}$ | $54.27_{\pm 0.014}$ | $52.03_{\pm 0.001}$ | $48.08_{\pm 0.004}$ | $58.65_{\pm 0.006}$ | $59.10_{\pm 0.025}$ |

Comparing option **A** to **C**, the results of Table 12 show that complementing the cross-entropy loss with the margin-based loss increased model performance by 0.38% in FGSM accuracy, 0.95% in PGD-20, 0.85% in PGD-100, 0.39% in CW, 2.71% in SQUARE, and 0.53% in AA. Confirming the benefit of using margin-based loss to generate the worst-case samples, leading to more robust models. In addition, Comparing option **B** to **C**, under both cases, we complemented the cross entropy loss with the margin-based loss to generate the worst-case adversarial sample used for training. The results of Table 12 show that complementing the cross-entropy loss with the margin-based in the outer minimization (loss used for training) increased model performance by 0.96% in natural accuracy, 0.83% in FGSM, 0.47% in PGD-20, 1.85% in SQUARE, and 0.43% in AA.

Comparing option C to option D, we observe that minimizing the disparity between the adversarial and natural moving averages of logits via the mHuber loss improves adversarial accuracy, with only a minor drop in natural accuracy. For example, we see a decrease in natural accuracy by 0.85% but an increase of 1.44% in PGD-20, 1.37% in PGD-100, 1.0% in AA, 0.89% in SQUARE, and 1.34% SPSA. The moving average of logits captures valuable information from previous iterations, which can enhance model robustness. Minimizing the disparity between natural and adversarial moving averages of logits allows for consideration of past disparities, making the process more efficient.

To further elaborate on the individual contribution of the margin loss to model robustness, we augmented the cross-entropy loss with the margin loss, omitting the mHuber regularization and the moving average of logits. The findings are documented in Table 14 below.

Table 13: This table provides an overview of the enhanced versions of the baseline losses (Standard AT, TRADES, and MART). The terms highlighted in bold represent the improvement strategies incorporated.

| Method | Improved Losses |
|---|---|
| Standard AT+$\boldsymbol{L'_i}$ | $\boldsymbol{L'_i} + CE(p(x'_i, \theta), y_i)$ |
| TRADES+$\boldsymbol{L'_i}$ | $\boldsymbol{L'_i} + CE(p(x_i, \theta), y_i) + \frac{1}{\lambda}KL(p(x_i, \theta)||p(x'_i, \theta))$ |
| MART+$\boldsymbol{L'_i}$ | $\boldsymbol{L'_i} + BCE(p(x'_i, \theta), y_i) + \lambda \cdot KL(p(x_i, \theta)||p(x'_i, \theta)) \cdot (1 - p_{y_i}(x_i, \theta))$ |

Table 14: Clean and robust accuracy on **ResNet-18** and Under **CIFAR-10**. We perform six runs and report the average performance with 95% confidence intervals. The 'Clean' column represents accuracy on natural examples.

| *Method* | Clean | PGD-20 | PGD-100 | CW | AA |
|---|---|---|---|---|---|
| vanillaAT | $85.80_{\pm 0.001}$ | $52.05_{\pm 0.003}$ | $49.28_{\pm 0.0022}$ | $51.08_{\pm 0.001}$ | $46.62_{\pm 0.004}$ |
| TRADES | $82.46_{\pm 0.0012}$ | $54.78_{\pm 0.0010}$ | $53.45_{\pm 0.0032}$ | $51.65_{\pm 0.0021}$ | $49.08_{\pm 0.0031}$ |
| MART | $81.30_{\pm 0.003}$ | $54.73_{\pm 0.006}$ | $53.28_{\pm 0.005}$ | $51.86_{\pm 0.0031}$ | $49.01_{\pm 0.0020}$ |
| **vanillaAT + L'$_i$** | $84.80_{\pm 0.003}$ | $54.79_{\pm 0.0012}$ | $53.32_{\pm 0.0023}$ | $51.96_{\pm 0.003}$ | $48.13_{\pm 0.001}$ |
| **TRADES + L'$_i$** | $84.15_{\pm 0.0013}$ | $56.03_{\pm 0.017}$ | $54.84_{\pm 0.0022}$ | $51.90_{\pm 0.021}$ | $49.16_{\pm 0.0034}$ |
| **MART + L'$_i$** | $82.65_{\pm 0.032}$ | $57.44_{\pm 0.012}$ | $56.00_{\pm 0.002}$ | $51.87_{\pm 0.012}$ | $48.90_{\pm 0.012}$ |

The results in Table 14 show that applying the margin loss to TRADES and MART significantly enhances model performance on both natural and adversarial examples, particularly under PGD-20 and PGD-100 while maintaining comparable accuracy on CW and AA. When the margin loss is applied to vanilla AT, we observe an improvement in adversarial robustness under PGD-20, PGD-100, CW, and AA, confirming the benefits of the margin loss in enhancing model robustness against adversarial attacks. This improvement is expected, as the margin loss is designed to push the decision boundary further from the training samples, making it harder for adversarial perturbations to cause misclassification, as more significant perturbations would be required for the sample to cross the decision boundary.

## 7 Conclusion

This paper introduces an enhancement strategy addressing scientists' concerns regarding deep learning models' vulnerability. Our method involves augmenting the cross-entropy loss with a margin-based loss to bolster the model's resilience against adversarial inputs. Furthermore, we introduce a novel training objective termed Logits Moving Average Adversarial Training (LMA-AT), which leverages the moving average of logits to regularize our model training process. Experimental results illustrate the effectiveness of our approach, achieving a better trade-off between natural accuracy and adversarial robustness than existing works. Integrating the margin loss with cross-entropy loss could provide a more robust defense against adversarial attacks. The margin loss encourages greater class separation, which can help the model resist adversarial perturbations, while cross-entropy loss ensures accurate classification. This combination may improve robustness without sacrificing too much natural accuracy. In addition, regularizing with the mHuber between the natural and adversarial moving averages of logits can stabilize the training by mitigating the impact of outliers and reducing the variance between natural and adversarial logits. This could lead to better generalization on both clean and adversarial data. Regarding scalability, the combined approach may scale well to large datasets and complex models, as the margin loss and cross-entropy losses can be adapted to various architectures. This adaptability can make it feasible to apply the technique to real-world problems involving large-scale data and models. Additionally, the method could be effective across various tasks and domains, such as image classification, natural language processing, and more. It may help develop more resilient models to maintain performance across different applications. When it comes to future work, we are interested in future research that could focus on optimizing the balance between the margin loss, cross-entropy loss, and mHuber regularization. Investigating how different values for these parameters affect model performance and robustness will be crucial.

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

# 8 Appendix A

## 8.1 Theoretical Justification for Information Retention in the Moving Average of logits

To theoretically illustrate that the Exponential Moving Average (EMA) of logits contains valuable information from previous iterations, we wil analyze the properties of the EMA and its ability to reflect historical information and trends in a time series of logits. Let $logit_t$ be the logit at epoch $t$, and $EMA_{t-1}$ represents the exponential moving average from the previous epoch. According to the definition of the moving average, The EMA is defined recursively as:

$$EMA_t = \tau \cdot logit_t + (1 - \tau) \cdot EMA_{t-1} \tag{18}$$

where $0 < \tau \leq 1$. We expand the recursive definition to show that this formula retains information from previous stages. According to the recursive definition of the moving average, the EMA at epoch $t$, $EMA_{t-1}$ is as follows:

$$EMA_{t-1} = \tau \cdot logit_{t-1} + (1 - \tau) \cdot EMA_{t-2} \tag{19}$$

Substitute equation (19) into the original equation (18) and we get:

$$EMA_t = \tau \cdot logit_t + (1 - \tau)\left(\tau \cdot logit_{t-1} + (1 - \tau) \cdot EMA_{t-2}\right) \tag{20}$$

Further simplification of the expression results in the following:

$$EMA_t = \tau \cdot logit_t + \tau \cdot (1 - \tau) \cdot logit_{t-1} + (1 - \tau)^2 \cdot EMA_{t-2} \tag{21}$$

We continue this expansion for $EMA_{t-2}$:

$$EMA_{t-2} = \tau \cdot logit_{t-2} + (1 - \tau) \cdot EMA_{t-3} \tag{22}$$

Substituting equation (22) into (21) and expanding yields

$$EMA_t = \tau \cdot logit_t + \tau \cdot (1 - \tau) \cdot logit_{t-1} + \tau \cdot (1 - \tau)^2 \cdot logit_{t-2} + (1 - \tau)^3 \cdot EMA_{t-3} \tag{23}$$

Generalizing, we get:

$$EMA_t = \sum_{i=0}^{t} \tau \cdot (1-\tau)^i \cdot logit_{t-i} \tag{24}$$

This series shows that the EMA at epoch $t$ is a weighted sum of all previous logits $logit_k$ (for $k = 0, 1, \ldots, t$), with the weights $\tau \cdot (1-\tau)^i$ decreasing exponentially as we go further back in time. The weights ensure that while recent logits significantly influence the current EMA, information from earlier logits is also retained, although with progressively lesser impact. Therefore, the EMA retains information from all previous stages, with the degree of retention controlled by the smoothing factor $\tau$.

## 8.2 Theoretical Justification that adversarial training of deep neural networks with margin loss and cross-entropy improves adversarial robustness

To demonstrate that combining margin loss with cross-entropy loss improves adversarial robustness, we break down the key components of the problem and illustrate how these training techniques enhance resistance to adversarial attacks.
Let $P(x)$ represent the probability density function of the input images, describing the likelihood of different images appearing in the dataset. Denote the output of a deep neural network for a given image $x$ as $f(x)$, and let $y$ be the true class label. Now, consider an adversarial perturbation $\epsilon$, which is a small change added to the input image $x$, resulting in a perturbed image $x' = x + \epsilon$. This perturbation is specifically crafted to cause the model to misclassify $x'$, i.e., $f(x') \neq y$. The objective is to minimize the probability of misclassification under adversarial perturbation with noise $\epsilon$.

### 8.2.1 Adversarial robustness in the context of cross-entropy loss

For a given image $x$ and true label $y$, the cross-entropy loss is defined as:

$$\mathcal{L}_{\text{CE}}(f(x), y) = -\log P(y|x) \tag{25}$$

where $P(y|x)$ is the predicted probability that the model assigns to the true label $y$ given the input image $x$. In the context of adversarial robustness, the goal is to understand how this loss function behaves when the input image $x$ is subjected to adversarial perturbations $\epsilon$. If $x' = x + \epsilon$ is the perturbed image, the cross-entropy loss for $x'$ would be:

$$\mathcal{L}_{\text{CE}}(f(x'), y) = -\log P(y|x') \tag{26}$$

Adversarial robustness aims to ensure that this loss does not significantly increase under perturbation, meaning that the model's confidence in the true label $y$ remains high even when the input is adversarially modified. The Cross-entropy loss encourages the model to maximize $P(y \mid x)$, making the model more confident in its predictions. The change in the model's output when a small perturbation $\epsilon$ is applied can be approximated by:

$$P(y \mid x + \epsilon) \approx P(y \mid x) + \nabla_x P(y \mid x) \cdot \epsilon. \tag{27}$$

during the training process, the Cross-entropy minimizes the gradient $\nabla_x P(y \mid x)$, making the model less sensitive to small perturbations, which improves robustness.

### 8.2.2 Adversarial Robustness in the Context of the Margin Loss

Consider the margin $\delta(x)$, which is the distance from the input image $x$ to the decision boundary. A larger margin means that $x$ is farther from the boundary, making it less likely that a small perturbation will result in misclassification. The Margin loss is designed to increase $\delta(x)$, effectively pushing the decision boundary away from the training samples. For a specific input $x$, the probability that a perturbation $\epsilon$ will cause a misclassification depends on whether the perturbation is large enough to push $x$ across the decision boundary. Formally, this probability is given by:

$$P(\text{misclass} \mid x, \epsilon) = P(\|\epsilon\| \geq \delta(x)). \tag{28}$$

Increasing the margin $\delta(x)$ decreases the likelihood that a small perturbation $\epsilon$ will cause misclassification, thereby enhancing robustness.

### 8.2.3 Joint Impact of Cross-Entropy and Margin Loss on Model Robustness

The Cross-entropy loss increases the model's confidence, reducing the sensitivity to small perturbations $\epsilon$. While the Margin loss increases the margin $\delta(x)$, reducing the probability that a perturbation $\epsilon$ will cross the decision boundary. Integrating Over All Possible Inputs $x$, To determine the overall probability of misclassification $P(\text{misclass} \mid x)$, we need to account for all possible inputs $x$ according to their distribution $P(x)$. This requires integrating the misclassification probability over all inputs. Hence, The overall probability of misclassification under an adversarial perturbation $\epsilon$ can be expressed as:

$$P(\text{misclass}, \epsilon) = \int_x P(\text{misclass} \mid x, \epsilon) P(x)\, dx = \int_x P(\|\epsilon\| \geq \delta(x)) P(x)\, dx, \tag{29}$$

Now, let's analyze the joint effect. Using the margin loss helps us to increase the margin $\delta(x)$. The region where $\|\epsilon\| \geq \delta(x)$ shrinks, reducing the misclassification probability. The Cross-entropy reduces $P(\text{misclass} \mid x, \epsilon)$ by making the model less sensitive to perturbations, further decreasing the overall misclassification probability. Together, these losses in adversarial training improve the adversarial robustness of deep neural networks in image classification by minimizing the probability of misclassification under adversarial perturbations.

## 9    Appendix B

In this section, we conducted additional experiments on benchmark datasets (CIFAR-10 and TinyImageNet) to evaluate the impact of the regularization parameter $\beta$ and the mHuber parameter $\alpha$ on the proposed LMA-AT loss. All experiments were performed using ResNet18 with a learning rate of 0.01, stochastic gradient descent (SGD) optimization with a momentum of 0.9, and a weight decay of 3.5e-3. Adversarial data used in training were generated using PGD with a random start, a maximum perturbation $\epsilon$ set to 8/255, a step size of 2/255, and the number of steps is 10, consistent with the settings described in Section 5.1.

**Sensitivity of the mHuber Hyperparameter $\alpha$:** A series of experiments were carried out to evaluate the sensitivity of the hyperparameter $\tau$. Starting with an initial value of 1.345, as recommended by (Huber, 2004) for the Huber function, we incremented $\alpha$ by 1 in each subsequent experiment.

Table 15: Assessing performance across various values of our modified Huber parameter, $\alpha$, under **CIFAR-10 with ResNet18** architecture.

| $\alpha$ | Natural | PGD-20 | PGD-100 | CW | SPSA | AA |
|---|---|---|---|---|---|---|
| 1.345 | $83.66_{\pm 0.001}$ | $56.79_{\pm 0.001}$ | $54.99_{\pm 0.002}$ | $52.29_{\pm 0.003}$ | $60.61_{\pm 0.012}$ | $48.31_{\pm 0.003}$ |
| 2.345 | $84.03_{\pm 0.004}$ | $57.17_{\pm 0.011}$ | $55.70_{\pm 0.001}$ | $52.52_{\pm 0.023}$ | $60.56_{\pm 0.005}$ | $48.80_{\pm 0.021}$ |
| 3.345 | $83.45_{\pm 0.0011}$ | $56.93_{\pm 0.003}$ | $54.99_{\pm 0.0010}$ | $52.44_{\pm 0.021}$ | $60.45_{\pm 0.002}$ | $48.57_{\pm 0.003}$ |
| 4.345 | $83.69_{\pm 0.002}$ | $57.03_{\pm 0.004}$ | $55.35_{\pm 0.001}$ | $52.12_{\pm 0.010}$ | $60.46_{\pm 0.001}$ | $48.44_{\pm 0.002}$ |
| **5.345** | $83.56_{\pm 0.0021}$ | $57.21_{\pm 0.001}$ | $55.64_{\pm 0.012}$ | $52.30_{\pm 0.001}$ | $60.44_{\pm 0.001}$ | $49.10_{\pm 0.001}$ |
| 6.345 | $84.30_{\pm 0.002}$ | $56.38_{\pm 0.002}$ | $54.48_{\pm 0.002}$ | $51.99_{\pm 0.002}$ | $61.13_{\pm 0.011}$ | $48.18_{\pm 0.002}$ |
| 7.345 | $84.27_{\pm 0.011}$ | $56.28_{\pm 0.003}$ | $54.51_{\pm 0.011}$ | $51.97_{\pm 0.003}$ | $60.16_{\pm 0.002}$ | $48.58_{\pm 0.003}$ |

In Table 15, comparing the model performance with $\alpha = 1.345$ to that with $\alpha = 5.345$, we observe an improvement in adversarial robustness. However, the robustness declines as $\alpha$ increases from 5.345 to 7.345, suggesting that when regularizing the adversarial training loss with the Modified Huber loss between natural and adversarial logits, the parameter $\alpha$ is critical in balancing the sensitivity of the loss function to small versus large errors, thereby directly influencing the model's robustness. When $\alpha$ is small, the Modified Huber loss becomes highly sensitive to minor differences between the moving averages of natural and adversarial logits. This heightened sensitivity can lead to more aggressive penalization of small adversarial perturbations,

prompting the model to focus on reducing even minor deviations. Conversely, larger values of $\alpha$ reduce this sensitivity, causing the loss function to overlook small logit differences and concentrate on more significant adversarial discrepancies. While this shift may enhance the model's defense against stronger attacks, it could also weaken its robustness against subtle perturbations. Therefore, a carefully chosen $\alpha$ can help the model resist subtle and strong adversarial attacks, improving the overall robustness.

**Sensitivity of the regularization parameter** $\beta$**:** We evaluate the impact of the regularization parameter $\beta$ on our proposed method on TinyImageNet, ResNet-18.

Table 16: Clean and robust accuracies on **TinyImageNet**, **ResNet-18** . We perform six runs and report the average performance with 95% confidence intervals. The 'Clean' column represents accuracy on natural examples.

| $\beta$ | Clean | PGD-20 | CW |
|---|---|---|---|
| 90 | $49.44_{\pm 0.003}$ | $25.17_{\pm 0.0024}$ | $21.50_{\pm 0.0014}$ |
| 92 | $49.26_{\pm 0.005}$ | $26.10_{\pm 0.004}$ | $22.10_{\pm 0.003}$ |
| 94 | $49.23_{\pm 0.0011}$ | $25.77_{\pm 0.0023}$ | $22.10_{\pm 0.0021}$ |
| 96 | $49.10_{\pm 0.001}$ | $26.35_{\pm 0.003}$ | $22.40_{\pm 0.006}$ |
| 98 | $48.79_{\pm 0.003}$ | $26.13_{\pm 0.013}$ | $22.27_{\pm 0.002}$ |

