# OpenReview forum: "Augmenting cross-entropy with margin loss and applying moving average logits regularization to enhance adversarial robustness"
_TMLR — Rejected by TMLR_

### Review · Reviewer_zPwy · 2024-07-28

**Summary Of Contributions:**

This paper proposes to combine a margin-based loss with the conventional cross-entropy loss, and to add a regularisation between adversarial and natural moving averages of logits, to improve adversarial robustness.

**Audience:**

Yes

**Claims And Evidence:**

No

**Requested Changes:**

For revision, please refer to Weakness 1, 2, 3, 4, 5, 7.

Response to Weakness 6 will be helpful.

**Strengths And Weaknesses:**

Strengths:
1. The proposed method is simple and effective. The idea is interesting and intuitive.
2. The experiments show that the robustness is improved.

Weaknesses:
1. In the introduction, the motivation is not stated very clearly. For example, what is the problem of the phenomenon in Fig.1 and what is the expected?
2. In Section 3, adding some intuitive and textual explanation for the mHuber loss will be helpful.
3. The proposed method is intuitive, but lacks a theoretical analysis and justification.
4. The hyper-parameters seems carefully tuned (alpha=6.345, 5.345, 2.345, beta=96, 86 etc.). How were them decided? Sensitivity experiments should be added.
5. In table 2 and table 3, the choices of tau are 0.0, 0.1, 0.2, 0.7, 0.9. How were these values selected?
6. In the experiments, the clean accuracy is not very good. The proposed method achieves better robustness by trading-off clean accuracy?
7. In table 9, more options are needed, for example, by removing the mHuber loss.

---

> ### Author Response · Authors · 2024-09-06
> **We appreciate the reviewer's valuable feedback. We have addressed the requested changes in the revised manuscript. For ease of reference, major changes have been highlighted in blue in the updated text.**
>
> 1- **In the introduction, the motivation is not stated very clearly. For example, what is the problem of the phenomenon in Fig.1, and what is the expected?**
>
>  When subjected to an attack, a model trained to recognize a jaguar may easily misclassify it as a cheetah or leopard, since these species share many similar features. Consequently, cheetah and leopard are the most likely incorrect classifications. This issue arises due to the closeness in predicted class probabilities. Our improvement strategy addresses this by widening the probability gaps between the correct class and the false ones, thereby enhancing the model's robustness. The change is reflected in the introduction. Thank you!
>
> 2- **In Section 3, adding some intuitive and textual explanation for the mHuber loss will be helpful.**
>
> Thank you for your valuable feedback. The requested changes have been reflected in the updated manuscript.
>
> 3- **The proposed method is intuitive, but lacks a theoretical analysis and justification.**
>
> We have added Appendix A to provide some theoretical justification.
>
> 4- **The hyper-parameters seems carefully tuned (alpha=6.345, 5.345, 2.345, beta=96, 86 etc.). How were them decided? Sensitivity experiments should be added.**
>
> We have included a description of how these parameters are selected in Section 5.1 Training Settings. Additionally, the updated manuscript now contains sensitivity experiments for these parameters. In Section 5.3.2, we included the sensitivity analysis of the regularization hyperparameter beta for ResNet18 on CIFAR-10, and CIFAR-100. The sensitivity on TinyImageNet is detailed in Appendix B. Additionally, Appendix B also covers the sensitivity analysis of the hyperparameter alpha for ResNet18 on CIFAR-10.
>
> 5-**In table 2 and table 3, the choices of tau are 0.0, 0.1, 0.2, 0.7, 0.9. How were these values selected?**
>
> Section 5.1 (Training Settings) includes a description of how our best value for tau is selected. We then choose different values around our best option (tau = 0.2) to assess the impact of this parameter in the training process. Ideally, we would assess values at 0.1, 0.2, 0.3, 0.4, 0.5, 0.6, 0.7, 0.8, and 0.9. However, we skipped some values due to limitations in computational resources.
>
> 6- **In the experiments, the clean accuracy is not very good. The proposed method achieves better robustness by trading-off clean accuracy?**
>
> There is a trade-off between adversarial robustness and natural accuracy [1]. However, our method achieves a more favorable balance compared to existing approaches. Although vanilla AT attains higher clean accuracy, it does so at the cost of significantly lower adversarial accuracy. The results in Tables 6, 7, and 8 demonstrate that our method not only outperforms other adversarial training (AT) variants in clean accuracy but also surpasses existing methods in most attack scenarios (further details highlighted in blue after Table 8) Therefore, our approach demonstrates a more balanced performance overall.
>
> [1]Tsipras, D., Santurkar, S., Engstrom, L., Turner, A., and Madry, A. Robustness may be at odds with accuracy. In International Conference on Learning Representations, 2019. \\
>
> 7- **In table 9, more options are needed, for example, by removing the mHuber loss.**
>
> We conducted additional experiments, and the findings are reported in  Section 6 (Ablation Studies). In Tables 11 and 12, we added option D. In addition,  we added tables 13 and 14.

---

### Review · Reviewer_gK7g · 2024-08-14

**Summary Of Contributions:**

This paper addresses the challenge of improving adversarial robustness in deep neural networks. It introduces a method that enhances the cross-entropy loss with a margin-based loss to generate adversarial samples during training. The proposed method also includes a novel training objective called Logits Moving Average Adversarial Training (LMA-AT), which utilizes the exponential moving average of logits to regularize the training process. Experimental results show that this approach effectively balances the accuracy and adversarial robustness. As a result, the disparity between clean and adversarial accuracy is reduced.

**Audience:**

Yes

**Broader Impact Concerns:**

Please discuss the potential impact at large scale and future works derived from this paper.

**Claims And Evidence:**

Yes

**Requested Changes:**

1.	Please discuss more in detail what are the limitations of the related work and how these issues are addressed in the proposed method.
2.	Please discuss the proposed method more in detail, describing all the operations involved. Please distinguish clearly between what is novel and what is implemented based on existing methods. Please also explain all the design decisions made to develop it.
3.	Please describe in more detail the experimental setup and tool flow used to conduct the experiments. If possible, the source code should be shared in an online repository to allow reproducibility.
4.	In Sections 5 and 6, each table of results should have a list of observation points to discuss the observations derived from the results.

**Strengths And Weaknesses:**

Strengths

1.	The tackled problem is relevant to the community.
2.	The contributions are clear.
3.	The experimental results are better than related work.

Weaknesses

1.	The proposed method is not described in sufficient detail.
2.	The discussion of the results and conclusion are not described in sufficient detail.

---

> ### Author Response · Authors · 2024-09-06
> **We appreciate the reviewer’s valuable feedback. The requested changes have been addressed in the revised manuscript. For your convenience, major changes have been highlighted in blue within the updated text.**
>
> 1- **Please discuss more in detail what are the limitations of the related work and how these issues are addressed in the proposed method.**
>
> We have revised the manuscript to incorporate the requested changes in Section 2.2 (Adversarial Defenses), with the updates highlighted in blue.
>
> 2- **Please discuss the proposed method more in detail, describing all the operations involved. Please distinguish clearly between what is novel and what is implemented based on existing methods. Please also explain all the design decisions made to develop it.**
>
> Thank you for your valuable feedback. We have included additional details in the updated manuscript, specifically in Sections 3 and 4.2, and particularly at the end of Section 4.4 (summary). The new text is highlighted in blue.
>
> 3- **Please describe in more detail the experimental setup and tool flow used to conduct the experiments. If possible, the source code should be shared in an online repository to allow reproducibility.**
>
> Thank you for your suggestion! Additional details are added in blue in the experiment section (Sections: 5.1 and 5.2); we will share the source code in an online repository as suggested.
>
> 4- **In Sections 5 and 6, each table of results should have a list of observation points to discuss the observations derived from the results.**
>
> The request has been addressed, and the changes are reflected in the updated manuscript.
>
> **Broader Impact Concerns:**
>
> **Please discuss the potential impact at large scale and future works derived from this paper.**
>
> As requested, we have updated the manuscript conclusion to incorporate your feedback.

---

### Review · Reviewer_MpTP · 2024-08-22

**Summary Of Contributions:**

This paper aims to enhance the adversarial robustness by proposing a variant of adversarial training. The proposed method increases the margin between the the ground-truth label's score and the scores of other classes. Specifically, it introduces a margin threshold to ensure that the score of the ground truth category for an adversarial example is higher than the scores of all other categories by at least this marginal threshold during adversarial training.

**Audience:**

Yes

**Broader Impact Concerns:**

No concerns.

**Claims And Evidence:**

No

**Requested Changes:**

See weaknesses.

**Strengths And Weaknesses:**

Strengths:

To enhance the adversarial robustness of adversarial training, this paper considers the effect of categories other than the ground-truth category on the classification results.

Weaknesses:

1.	The core contribution of this paper is Eq. (13), which aims to increase the discrepancy between the score of an adversarial example for the ground-truth label and the scores for other categories during adversarial training. However, in Table 1 and the experimental results, the authors did not directly evaluate the impact of this loss on standard methods such as AT, TRADES, and MART, both with and without the proposed loss. Instead, the authors evaluated the proposed loss in combination with the mHuber loss presented in [1]. Therefore, the experimental results do not clearly demonstrate the effectiveness of the proposed loss alone on standard methods. Furthermore, the effect of using the mHuber loss alone on standard methods, with or without the mHuber loss, is not reported.

2.	Table 4 shows that, under AutoAttack, the robustness of the proposed 'TRADES + Ours' and 'MART + Ours' is worse than that of TRADES and MART, respectively. Additionally, the robustness of the proposed method is inferior to that of PMHR-AT. Similarly, Table 5 shows that the robustness of the proposed method is also worse than PMHR-AT under AutoAttack.

3.	The experimental results are sensitive to hyperparameters. In Table 2, under AutoAttack, the results are weaker than TRADES, MART, and PMHR-AT when the moving average parameter $\tau$ = 0,0.1,0.7. Besides, in the experimental setup of Section 5.1, the parameter $\alpha$ of mHuber loss is set to 6.345 on TinyImageNet, 5.345 on CIFAR-10, and CIFAR-100 on ResNet-18, and 2.345 on WRN-34-10. The weight decay of the training is set to 3.5e-3. Please provide an explanation for the choice of these specific parameters.

Minor:

1. Should the Huber losses in Equations (15), (17), and (18) be denoted as $mHuber(\text{logit}_t', \text{logit}_t, \alpha)$?

2. Please clearly describe the physical significance of mHuber loss on page 5, including the roles of the vector $c$ and the matrices $A$ and $B$.

3. Eq.(10) is better presented as an example rather than as a formula.

4. Eq.(6) focuses on the top $q$ most likely incorrect categories. How is $q$ determined in the specific experiments?

[1] Atsague et al. A penalized modified huber regularization to improve adversarial robustness. In ICIP, 2023.

---

> ### Author Response · Authors · 2024-09-06
> **We appreciate the reviewer’s valuable feedback. We have made the requested changes in the revised version. Major changes in the text have been highlighted in blue to facilitate reference.**
>
> **Response to Comment 1**
>
> Thank you for your valuable feedback. We have added additional experiments in Section 6 (Ablation Studies). For example, Table 14 evaluates the individual contribution of the margin loss on standard methods AT, TRADES, and MART. Additionally, Tables 11 and 12 under option "D" assess the contribution of the mHuber regularization applied to the moving average of logits. These results in Table 14 show that applying the margin loss to TRADES and MART significantly enhances
> model performance on both natural and adversarial examples, particularly under PGD-20 and PGD-100 while
> maintaining comparable accuracy on CW and AA. When the margin loss is applied to vanilla AT, we observe
> an improvement in adversarial robustness under PGD-20, PGD-100, CW, and AA, confirming the benefits
> of the margin loss in enhancing model robustness against adversarial attacks.
>
> **Response to Comment 2**
>
> Thank you for your valuable feedback. As you pointed out, the robustness of the proposed 'TRADES + Ours' and 'MART + Ours' methods is slightly lower under AutoAttack than TRADES and MART. However, a closer examination reveals that TRADES and MART achieve this robustness at the expense of natural accuracy. In contrast, 'TRADES + Ours' and 'MART + Ours' demonstrate better robust accuracy under FGSM, PGD-20, PGD-100, and stronger Black Box attacks such as SQUARE and SPSA. While PMHR-AT exhibits strong robustness under AutoAttack, it does so at the cost of natural accuracy. Our method outperforms PMHR-AT in terms of natural accuracy, as well as robustness under FGSM, PGD-20/100, and stronger Black Box attacks like SQUARE and SPSA. Overall, our proposed method improves adversarial robustness in certain attacks and offers a better trade-off between natural and adversarial accuracy than existing approaches. We have updated the manuscript to reflect this change, highlighted in blue, below Table 7.
>
> **Response to Comment 3**
>
> Thank you for your feedback. Your observation is correct. However, our best performance was achieved with $\tau = 0.2$, where we outperformed the TRADES, MART, and PMHR-AT baselines in natural accuracy and in most attacks. Since this parameter determines how much past information is considered, improper calibration could significantly affect model performance. Additionally, as requested, we have provided further details on how the parameters were selected in the updated manuscript, specifically in Section 5.1 (Training Settings).
>
> **Minor:**
>
> 1-**Should the Huber losses in Equations (15), (17), and (18) be denoted as $mHuber(\text{logit}_t', \text{logit}_t, \alpha)$?**
>
> Yes, Equations (15), (17), and (18) have been adjusted accordingly in the revised version. Thank you!
>
> 2-**Please clearly describe the physical significance of mHuber loss on page 5, including the roles of the vector $c$ and the matrices $A$ and $B$.**
>
> The paper has been updated to reflect the requested changes in Section 3 (Notation and Preliminaries), with the revisions highlighted in blue.
>
> 3- **Eq.(10) is better presented as an example rather than as a formula.**
>
> We have followed your suggestion and updated the equation as an example in the revised paper.
>
> 4- **Eq.(6) focuses on the top $q$ most likely incorrect categories. How is $q$ determined in the specific experiments?**
>
> Interesting question! Technically, $q$ is not a fixed number. Depending on the classification task, if the classes are inherently very similar or not well-separated, $q$ may be closer to the total number of classes. We determine $q$ based on the proportion of data points that are incorrectly classified into a specific class. in this case, $q=5$

---

### Decision · Action_Editor_f8rp · 2024-10-18

**Recommendation:** Reject

**Comment:**

This paper presents a novel adversarial training method aimed at improving adversarial robustness, with experiments conducted on CIFAR-10, CIFAR-100, and TinyImageNet. However, the reviewers raised concerns regarding the effectiveness of the proposed method, particularly due to its unstable performance (The performance is sometimes inferior to TRADES and MART) and the need for more carefully tuned parameters. After the discussion, these concerns were not fully addressed in the authors' response, leading one reviewer to maintain a "Reject" recommendation. Overall, the manuscript is not yet ready for publication.

**Audience:**

Due to the effectiveness of the method has not been demonstrated, the interested individuals will be limited.

**Claims And Evidence:**

The effectiveness of the proposed method is not well demonstrated by experimental results. The proposed methods are sometimes inferior the competing method and the results also depend on well-designed parameters.

**Resubmission Of Major Revision:**

The authors may consider submitting a major revision at a later time.